# Sculpting Latent Spaces With MMD: Disentanglement With Programmable Priors

## Abstract

Learning disentangled representations, where semantic features are captured by independent variables, is dominated by the Variational Autoencoder (VAE) which uses the Kullback-Leibler (KL) penalty to learn a factorized representation in the latent space. In this paper, we provide direct visual and quantitative evidence that the VAE-based methods consistently fail to enforce this target distribution on the aggregate posterior, subsequently falling short of a mutually independent representation – the holy grail objective of unsupervised disentanglement. We quantify this failure and resulting entanglement using a stable, unsupervised Latent Predictability Score (LPS). To address this, we propose the Programmable Prior Framework: a non-parametric method built on the Maximum Mean Discrepancy (MMD). We verify our framework allows practitioners to explicitly sculpt the latent space, achieving (1) state-of-the-art unsupervised statistical independence (measured by LPS), (2) alignment to semantic features using an internal semi-supervised mechanism, and (3) aggregate posterior distribution shaping (validated through quantization-aware training), all without reconstruction trade-offs. Ultimately, the framework is one of a kind in that it provides a reliable foundational tool for balancing these three key training objectives, opening new avenues for model identifiability, interpretability, causal reasoning, and efficient compression.

## 1 Introduction

A particularly promising goal of modern machine learning is to learn *disentangled* representations, where individual latent units are mutually independent, and semantically interpretable (Higgins et al., 2017). Achieving this is considered a critical step towards more robust and generalizable models, ultimately enabling capabilities like compositional generalization, fairness, and causal reasoning (Schölkopf et al., 2021). However, disentangled representations are not unique and guaranteed alignment to specific semantic factors is understood to necessitate some level of supervision Hyvärinen & Pajunen (1999); Khemakhem et al. (2020); Locatello et al. (2019; 2020).

The dominant paradigm for disentanglement, established by a sequence of seminal contributions (Higgins et al., 2017; Kim & Mnih, 2018; Chen et al., 2018; Burgess et al., 2018; Mathieu et al., 2019; Locatello et al., 2020; Khemakhem et al., 2020; Locatello et al., 2020; Khemakhem et al., 2020), has been to use a Variational Autoencoder (VAE) framework, which uses a stochastic reparametrization trick and the Kullback-Leibler (KL), to constrain the latent space's posterior distribution. However, this approach has been scrutinized for failing to achieve true disentanglement (Locatello et al., 2019), as the recovery of interpretable factors is often an incidental byproduct of random initialization rather than a guaranteed outcome of the objective, highlighting the fundamental unidentifiability of the unsupervised setting, and for its architectural rigidity, tying the disentanglement objective to the specific structure of a VAE.

In this work, we decouple the broad objective of disentanglement into learning features that are both **(1) mutually independent (as in Nonlinear Independent Component Analysis or NICA)**, and **(2) semantically meaningful**. To realize this decoupling and address the shortcomings of prior work, we utilize a sculpting regularizer for the latent space's aggregate posterior distribution based on the architecture-agnostic and non-parametric Maximum Mean Discrepancy (MMD) (Gretton et al., 2012). Ultimately, our framework is one of a kind in its ability to program a model's latent space rep-

resentation to address three key representational properties: mutual independence, semantic meaningfulness, and Latent Distribution Shaping. Our primary contributions are summarized as follows:

- **A Critical Failure Analysis of VAE-based Disentanglement.** We provide direct visual (Fig 1) and quantitative evidence (Table 1) that the dominant VAE-based framework (including $\beta$-VAE, FactorVAE, and $\beta$-TCVAE) fails to achieve its stated goal of enforcing a factorized prior. We also show that the parametric KL-divergence is an unreliable mechanism, leading to demonstrably entangled representations (as verified by a high LPS score).

- **A Stable, Unsupervised Metric for Mutual Independence.** We introduce the Latent Predictability Score (LPS), closely related to the training objective of the NashAE Yeats et al. (2022), as an unsupervised metric for quantifying mutual independence. We frame it as a necessary "rejection test": a high LPS score (high predictability) allows us to reliably reject the hypothesis that a representation is disentangled. We demonstrate in Appendix C that LPS is exceptionally stable, in stark contrast to high-variance, supervised metrics like DCI and MIG.

- **A Unified Framework for Independence, Alignment, and Distribution Shaping.** We introduce the Programmable Prior Framework, a non parametric, architecture-agnostic MMD-based sculpting regularizer that unifies and balances three distinct training goals:
  1. **Statistical Independence (1):** It successfully sculpts the latent space to match a factorized prior, achieving state-of-the-art independence (SOTA LPS) where VAE-based methods fail.
  2. **Semantic Alignment (2):** It provides a semi-supervised mechanism to mitigate the NICA unidentifiability problem by aligning latents to *pseudo-labels*.
  3. **Distribution Shaping (3):** To validate the efficacy of our framework for aggregate posterior shaping, we highlight its usage through the application of quantization aware training (QAT) for data compression. The effectiveness of the latent quantizer hinges on the aggregate posterior matching the quantizer's expected distribution.

We further demonstrate that our framework achieves all three goals without the reconstruction-quality trade-off that plagues VAE-based methods.

To facilitate reproducibility and encourage further research, our code is made publicly available.[1]

## 2 BACKGROUND AND PRIOR WORK

This section provides the necessary background on the generative models, theoretical challenges, and statistical tools that form the context for our work. We begin by formalizing the autoencoder and variational autoencoder paradigms before discussing the core challenges of disentanglement and introducing the Maximum Mean Discrepancy as a powerful non-parametric regularizer.

**Latent Variable Models and the Variational Autoencoder.** In its simplest form, a latent variable model is defined by a joint distribution $p(x, z)$ of observed data $x$ and latent variables $z$. In deep learning, this relationship is parameterized by a neural network, often subdivided into an encoder (or inference model) $q_{\theta_1}(z|x)$ and a decoder (or generative model) $q_{\theta_2}(x|z)$. A natural training objective is to maximize the log-likelihood, $\mathbb{E}_{p(x)}[\log q_\theta(x)]$. Training these two components end-to-end with this objective is the basis of the classical Autoencoder (AE) paradigm.

Prior work has largely focused on frameworks that provide the ability to specify and constrain the latent space's posterior distribution $p(z|x)$. Re-writing the log-likelihood as $\mathbb{E}_{p(x)}[\log \mathbb{E}_{p(z)}[q_\theta(x|z)]]$ provides the most obvious objective to maximize, but direct optimization in this form is generally intractable. As a solution, the Variational Autoencoder (VAE) framework provides a popular and effective solution that consists in maximizing the Evidence Lower Bound (ELBO) of the log-likelihood:

$$\mathcal{L}_{\text{ELBO}}(x) := \underbrace{\mathbb{E}_{q_{\theta_1}(z|x)}[\log q_{\theta_2}(x|z)]}_{\text{Reconstruction Loss}} - \underbrace{D_{\text{KL}}(q_{\theta_1}(z|x)||p(z))}_{\text{Regularization Loss}} \quad (1)$$

---

[1]Code url: `https://github.com/my-anonymous-repo/sculpting-latent-spaces`

In a standard VAE, the encoder outputs the parameters of a diagonal Gaussian, a mean vector $\mu(x)$ and a variance vector $\sigma^2(x)$. A latent code is then sampled via the reparameterization trick: $z = \mu(x) + \sigma(x) \cdot \epsilon$, where $\epsilon \sim \mathcal{N}(0, I)$. The regularization term becomes the per-sample KL-divergence between the approximate posterior and the prior, which is typically a standard normal distribution: $D_{\mathrm{KL}}(N(\mu(x), \sigma(x)) || N(0, I))$.

**The Challenge of Unsupervised Disentanglement.** In the context of unsupervised disentanglement, the overarching goal of the VAE-based framework is to learn a representation where the latent vector $z$ is factorized, meaning its components are statistically independent. The 'Regularization Loss' is thought to encourage this by forcing the posterior distribution towards a 0 mean identity covariance Gaussian. In fact, seminal works like $\beta$-VAE (Higgins et al., 2017), Factor-VAE (Kim & Mnih, 2018), and $\beta$-TCVAE (Chen et al., 2018) all build on this principle through distinct tweaks to the $D_{\mathrm{KL}}$ term. However, Figure 1 (a-b) demonstrates that this mechanism is unreliable as the VAE-based methods fail to enforce the desired distribution on the aggregate posterior $\mu(x)$: If $p(z|x) \sim N(0, I)$ as is the objective of the VAE framework, then so should the aggregate $p(z) = \mathbb{E}_p(x)[p(z|x)] \sim N(0, I)$. The non-identity covariance matrix demonstrates substantial entanglement while the non bell-curved marginals prove the learned representations cannot possibly be Gaussian.

Other, but less popular alternative paradigms exist, such as using adversarial training (Chen et al., 2016; Yeats et al., 2022) or imposing direct geometric constraints like the learning of orthogonal group action features (Cha & Thiyagalingam, 2023).

In the end, unsupervised learning of latent feature representations that are both (1) mutually independent, and (2) semantically meaningful is complicated by deep theoretical and practical challenges. The unidentifiability of NICA (Hyvärinen & Pajunen, 1999; Khemakhem et al., 2020), which posits that independent and interpretable representations are not unique, implies that some level of supervision is necessary for (2) (Locatello et al., 2019; 2020).

**The Maximum Mean Discrepancy.** An alternative to the KL-divergence for estimating the distance between two distributions is the Maximum Mean Discrepancy (MMD) (Gretton et al., 2012). The core intuition of MMD is to represent distributions as single mean embeddings in a high-dimensional Reproducing Kernel Hilbert Space (RKHS). The MMD between two distributions, $P$ and $Q$, is then simply the RKHS norm of the difference between their mean embeddings, $\mu_P$ and $\mu_Q$:

$$\mathrm{MMD}(P, Q) = \|\mu_P - \mu_Q\|_{\mathcal{H}} \tag{2}$$

If the mean embeddings are identical, the MMD is zero, and the distributions are identical. This is operationalized via a kernel function, $k(\cdot, \cdot)$, such as a Gaussian RBF kernel.

In practice, finite set of samples $Z = \{z_1, \dots, z_m\} \sim P$ and $Z' = \{z_1', \dots, z_n'\} \sim Q$ are used to estimate the mean embeddings. The MMD is then the distance between these empirical estimates:

$$\mathrm{MMD}(P, Q) \approx \left\| \frac{1}{m} \sum_{i=1}^{m} \phi(z_i) - \frac{1}{n} \sum_{j=1}^{n} \phi(z_j') \right\|_{\mathcal{H}} \tag{3}$$

where $\phi$ is the feature map into the RKHS associated with the kernel $k$. Using the kernel trick, $\langle \phi(a), \phi(b) \rangle_{\mathcal{H}} = k(a, b)$, we can expand this into a computable form. The standard unbiased empirical estimator for the squared MMD is given by the sum of three terms:

$$\mathrm{MMD}^2(P, Q) = \frac{1}{m(m-1)} \sum_{i \neq j} k(z_i, z_j) - \frac{2}{mn} \sum_{i,j} k(z_i, z_j') + \frac{1}{n(n-1)} \sum_{i \neq j} k(z_i', z_j') \tag{4}$$

The principle of using an MMD penalty as a more stable substitute for other divergence function in Neural Networks is a well-established concept, primarily explored for generative modeling, domain adaptation, and representational knowledge distillation (Tolstikhin et al., 2017; Li et al., 2017; Long et al., 2017; Huang & Wang, 2017; Zhao et al., 2019). While our work leverages a similar core mechanism as that of the Wasserstein Autoencoder (Tolstikhin et al., 2017), which explored MMD as a more stable divergence in the context of generative modeling, it is fundamentally distinguished by its objective. In the section that follows, we re-purpose this tool for a different goal entirely: **programmable disentanglement**. Our framework shifts the focus to **representational engineering**, using the MMD regularizer as a precise instrument to inject a wide range of explicit, user-defined inductive biases.

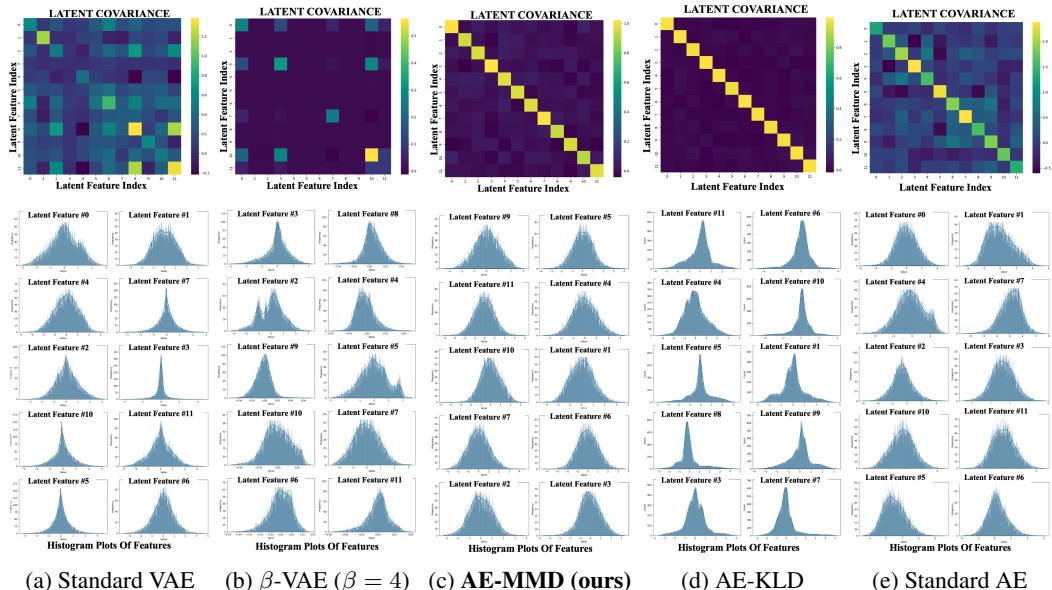

|  (a) Standard VAE | (b) $\beta$-VAE ($\beta = 4$) | (c) **AE-MMD (ours)** | (d) AE-KLD | (e) Standard AE |

Figure 1: **Visual comparison of learned latent distributions on MNIST for key baselines.** Each column corresponds to a different model, showing the covariance matrix between the latent space features over the dataset (top row) and the histogram plots for the marginal distribution of the latent space features over the whole dataset (bottom row). The VAE-based models (a, b) and the standard AE (e) fail to enforce either independence or the target Gaussian geometry. While a batch-wise KLD regularizer (d) achieves a diagonal covariance, its parametric nature prevents it from correctly shaping the marginal distributions leading to co-dependence (LPS equal to 0.97). Only our MMD-regularized model (c) successfully enforces both properties, producing a representation that is truly independent and matches the target prior.

## 3 THE MMD REGULARIZER FOR SCULPTING THE LATENT SPACE

Our core technical contribution is the formulation of an MMD-based sculpting framework (visualized in Figure 2) that can be integrated into any neural network layer where it is desired for the aggregate posterior, $z \sim q_{\theta_1}(z) = \mathbb{E}_{p(x)}[q_{\theta_1}(z|x)]$, to match a prior distribution $p(z)$. With this in mind, our framework proposes to maximize the following lower bound on the log-likelihood,

$$\mathcal{L}_{\text{ours}} := \underbrace{\mathbb{E}_{p(x)}[\log q_\theta(x)]}_{\text{Log-likelihood}} - \lambda \cdot \underbrace{\text{MMD}^2(q_{\theta_1}(z), p(z))}_{\text{MMD Regularization}}, \tag{5}$$

where $\lambda$ is a hyperparameter that controls the strength of the regularization. This is different from a VAE which attempts to enforce a target prior directly on the posterior distribution $p(z|x)$ for each $x$. This results in our objective being a standard regularized objective, not a VAE-style ELBO.

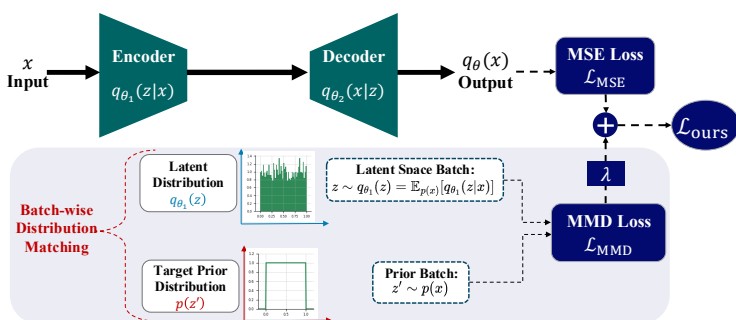

Figure 2: Visual Representation of the Programmable Prior Framework.

## 3.1 Disentanglement and the Programmable Prior

A central advantage of our framework is the ability to choose the target prior $p(z)$. By programming the prior, we can sculpt the aggregate posterior's geometry to have specific, desirable properties such as (1) mutual independence and (3) latent distribution shaping. But, we can also sculpt the aggregate posterior to have complex internal dependencies and (2) semantically meaningful features.

**The Failure of KL-based Regularization.** A more efficient approach than MMD is to apply the analytic KL divergence to the batch statistics of the encoded features, using a loss term like $D_{KL}(\mathcal{N}(\mu_z, \Sigma_z) || \mathcal{N}(0, I))$, where $\mu_z$ and $\Sigma_z$ are the empirical mean and covariance. This is equivalent to our framework in equation 5 if the MMD regularization were replaced with KLD regularization. While this correctly targets the aggregate distribution, its parametric nature is a critical flaw. As shown in Figure 1(d), this batch-wise KL regularizer successfully enforces a diagonal covariance but fails to make the marginals Gaussian leading to consequential entanglement. In fact, we report an LPS score of 0.97 for this method, indicating the features are completely co-dependent over the data. This demonstrates that merely matching the first two moments is insufficient for enforcing true statistical independence.

**A Solution For Sculpting The Latent Space.** This motivates the need for a non-parametric, sample-based regularizer. Switching from an analytical penalty like KLD to the MMD allows for far more flexible priors. Because MMD operates on samples, the target distribution does not need to be analytically defined; we only need a way to sample from it. This allows us to use fixed priors, like a Gaussian, or even adaptive priors derived from data during training. As shown in Figure 1(c), our MMD-based approach is the only method that successfully enforces both a perfectly diagonal covariance matrix and perfectly Gaussian marginal distributions, matching the entire distribution, not just its moments.

The true flexibility of our MMD framework is best demonstrated by its ability to enforce priors with complex, structured dependencies. To showcase this, we designed an experiment to push the limits of MMD regularization. First, we trained an unconstrained, standard Autoencoder and saved the dataset of its learned latent vectors. This collection of latent codes—with its complex, co-dependent structure—served as the target prior for a new model trained with our framework. This experiment serves as a demonstration of MMD's power to enforce a prior so intricate it cannot be written analytically for a KLD-based regularizer. The results in Figure 16 visually confirm that our method can convincingly copy this arbitrary, entangled latent geometry with high fidelity, highlighting the precision of MMD as a tool for representational engineering.

**A Solution For Semi-Supervised Alignment To Interpretable Factors.** While achieving unsupervised statistical independence (1) is satisfied with a factorized prior $p(z)$, the unidentifiability of NICA implies that independent representations are not necessarily semantically meaningful. To learn semantically meaningful features (2), we utilize a semi-supervised mechanism inside of our Programmable Prior Framework to inject the necessary **inductive bias**.

To integrate semantic features or **pseudo-labels** $u \sim p(u|x)$, we define a "programmable" target prior over the joint pairs $(z, u)$ that enforces a one-to-one correspondence. Specifically, we employ a lightweight auxiliary network, $f_\phi$ (e.g., an element-wise single-hidden-layer MLP), to map the semantic factors $u$ into the statistical domain of the latent space (e.g., transforming angular degrees into Gaussian values), thereby allowing for this desired relationship to be learned. This network is trained jointly with the encoder. The MMD-based sculpting objective is then updated to balance our three key training objectives through minimization of the discrepancy between the joint distributions:

$$\mathcal{L}_{\text{ours}} := \mathbb{E}_{p(x)}[\log q_\theta(x)] - \lambda \cdot \text{MMD}^2\Big(q_{\theta_1}(z, f_\phi(u)), p(z, f_\phi(u))\Big) \qquad (6)$$

By minimizing this joint MMD, the encoder and auxiliary networks are forced to learn a mapping $q_{\theta_1}(z|f(u))$ where specific latent dimensions are locked to specific semantic factors. Crucially, this is achieved with the same MMD-based sculpting regularizer defined in equation 5 with the exception that extra information is appended to each sample. This mechanism unifies three distinct goals—(1) statistical independence, (2) semantic alignment, and (3) latent distribution shaping—into a single, elegant framework, demonstrating the true power of a programmable prior for representation engineering.

# 4 EXPERIMENTS AND RESULTS

In this section, we first motivate the need for a new, unsupervised evaluation paradigm and formally introduce the Latent Predictability Score (LPS). Following this, we present a comprehensive analysis of our Programmable Prior Framework through three key experimental lenses: (1) enforcing mutual independence in the latent space, (2) achieving semantic alignment with interpretable factors, and (3) aggregate posterior distribution shaping demonstrated through compatibility with downstream quantization tasks.

## 4.1 A NOVEL UNSUPERVISED METRIC FOR LATENT INDEPENDENCE

**Motivation.** A significant limitation of popular disentanglement metrics like the mutual information gap (MIG) (Chen et al., 2018) or the Disentanglement Completeness Information score (DCI) (Eastwood & Williams, 2018) is their reliance on access to semantically meaningful factors for evaluation. Such labels are unavailable for most real-world datasets, creating a critical gap for a metric that can directly quantify the quality of a learned representation in a truly unsupervised manner. Moreover, several works at the interface of NICA and disentanglement have highlighted that **semantically meaningful and disentangled representations are not unique** which considerably degrades the reliability of any supervised metric Locatello et al. (2019; 2020); Khemakhem et al. (2020). To address these challenges, we propose to use an unsupervised metric that directly measures the level of mutual independence of the learned latent features over the data, directly quantifying the satisfiability of step (1) for obtaining disentangled representations.

**Introducing the Latent Predictability Score (LPS).** The core intuition is simple and closely related to the training objective of the NashAE (Yeats et al., 2022): if a $d$ dimensional latent space, with latent features $\{z_1, \ldots, z_d\}$ are truly independent, then any given feature $z_i$ should not be predictable from the others, $z_{j \neq i}$. The LPS procedure quantifies this intuition as follows:

1. A representative set of latent vectors is obtained by encoding a large sample of data.
2. For each latent dimension $i$ from 1 to $d$:
   (a) The feature $z_i$ is treated as a target label.
   (b) A regression model is trained to predict $z_i$ using the remaining $d-1$ dimensions, $z_{j \neq i}$.
   (c) The quality of this prediction is evaluated on a withheld test set using the coefficient of determination ($R^2$) score.
3. The final LPS is the average $R^2$ score across all $d$ dimensions.

**Discussion.** The final score is a direct measure of latent dependency. A higher score implies that the features are entangled and predictable from one another, while a score approaching zero indicates a high degree of mutual independence. Therefore, a **lower LPS score is better**. In Appendix C, we provide a thorough analysis of this LPS metric compared to classical supervised metric alternatives. We observe that the LPS metric is far more stable than the popular DCI score which has an overwhelming amount of variation in results for the same baselines. Logically, the LPS metric and supervised alignment metrics like DCI should be used hand in hand to evaluate the quality of disentangled representations in achieving (1) mutual independence and (2) semantic alignment respectively.

**Variants.** To ensure the predictability measure is robust, we employ two powerful and distinct regression models for this task. We therefore report two versions of our metric: **LPS-lgbm**, which uses the highly effective LightGBM gradient boosting framework (Ke et al., 2017) , and **LPS-mlp**, which uses a 2 hidden layers Multi-Layer Perceptron (reported in the appendices).

## 4.2 EXPERIMENTAL SETUP

**Datasets.** We evaluate our method across a diverse suite of benchmarks to test its scalability and effectiveness. We use synthetic datasets with known semantic factors: **dSprites** (Matthey et al., 2017), **CelebA** (Liu et al., 2015), and the simple generative environments **XY**, **XYC**, and **XYCS** introduced in (Cha & Thiyagalingam, 2023), which consist of circles with varying position (XY), color intensity (C), and shape (S). To test performance on complex, real-world data, we use **MNIST** (Deng, 2012), **CIFAR-10** (Krizhevsky et al., 2009), and **TinyImageNet** (Le & Yang, 2015).

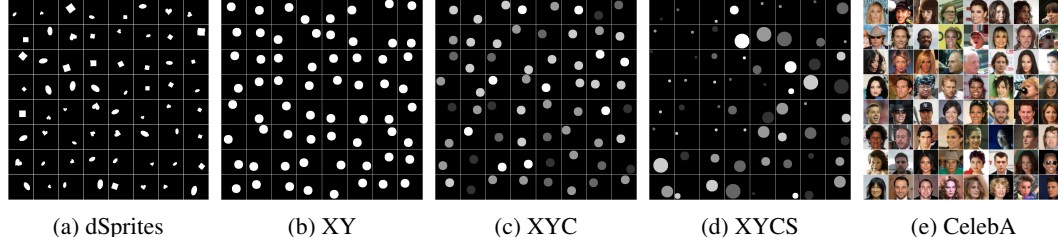

| (a) dSprites | (b) XY | (c) XYC | (d) XYCS | (e) CelebA |

Figure 3: Example images (represented in an $8 \times 8$ grid) from the synthetic datasets used in our experiments. From left to right: (a) dSprites, with factors like shape, scale, and orientation; (b-d) The XY family of datasets, with factors for position (XY), color (C), and shape (S); (c) The CelebA dataset consisting of faces with 40 different semantic factors.

**Baselines.** We compare our MMD-based approach against several established and state-of-the-art methods for unsupervised disentanglement. Our primary baselines are $\beta$-**VAE** (Higgins et al., 2017), $\beta$-TCVAE Chen et al. (2018), FactorVAE Kim & Mnih (2018), and **DGAE** (Cha & Thiyagalingam, 2023). We also include a standard **Autoencoder (AE)** with no explicit disentanglement regularizer and standard **VAE** as control variables to establish a performance floor.

While our reported experiments used a batch size of $512$ and a regularization coefficient of $\lambda = 0.3$, the MMD regularizer proved robust, yielding consistent results across a wide range of these hyperparameters. The implementation, comprehensive experimental results, and ablation studies for the MMD-based sculpting regularizer can be found in Appendix A, B and D respectively.

### 4.3 STATE-OF-THE-ART MUTUAL INDEPENDENCE

For unsupervised disentanglement, our quantitative analysis reveals that our MMD-based regularizer achieves state-of-the-art performance in enforcing latent statistical independence, a claim supported across a wide range of datasets and model configurations. The condensed results in Table 1 demonstrate that our method consistently yields the lowest Latent Predictability Scores (LPS), signifying a higher degree of mutual independence, with little to no degradation in reconstruction quality. To contextualize these scores, the LPS measures the proportion of a latent feature's variance predictable from its peers using the coefficient of determination ($R^2$). For instance, on dSprites, the popular $\beta$-VAE achieves an LPS of $0.52$ implying substantial entanglement; in contrast, our score of $0.04$ signifies a representation approaching true independence.

Table 1: Main quantitative results comparing our unsupervised MMD-based Autoencoder against baselines (the latent space dimension is specified next to the dataset name). Each cell presents two key metrics: **LPS-lgbm** ($R^2$) — *reconstruction* **NMSE (dB)** where lower values are better ($\downarrow$). The best performance for each metric on each dataset is highlighted in **bold**.

| Model | ImageNet ($d=64$) LPS | NMSE (dB) | CIFAR10 ($d=64$) LPS | NMSE (dB) | MNIST ($d=12$) LPS | NMSE (dB) | dsprites ($d=5$) LPS | NMSE (dB) | XY ($d=2$) LPS | NMSE (dB) | XYC ($d=3$) LPS | NMSE (dB) | XYCS ($d=4$) LPS | NMSE (dB) | celebA ($d=32$) LPS | NMSE (dB) |
|---|---|---|---|---|---|---|---|---|---|---|---|---|---|---|---|---|
| **AE-MMD (Ours)** | **0.03** | **-11.70** | **0.04** | **-15.61** | **0.22** | **-11.92** | **0.04** | **-17.89** | **-0.05** | -15.22 | **-0.01** | **-16.31** | **0.11** | **-16.76** | 0.08 | -14.37 |
| DGAE | 0.77 | -10.3 | 0.69 | -15.53 | 0.63 | -11.48 | 0.56 | -14.61 | 0.25 | -6.74 | 0.50 | -10.94 | 0.59 | -15.02 | 0.48 | **-14.40** |
| FactorVAE ($\gamma$=10.0) | 0.44 | -10.75 | 0.63 | -12.49 | 0.27 | -8.62 | 0.08 | -14.56 | -0.04 | -15.92 | 0.00 | -15.52 | 0.17 | -15.52 | **0.07** | -14.39 |
| $\beta$-TCVAE ($\beta$=4.0) | 0.79 | -9.80 | 0.89 | -10.68 | 0.84 | -5.97 | 0.22 | -8.83 | -0.04 | -14.68 | **-0.01** | -13.72 | 0.17 | -12.40 | 0.59 | -13.10 |
| $\beta$-VAE ($\beta$=4) | 0.73 | -9.87 | 0.85 | -10.49 | 0.90 | -4.61 | 0.52 | -8.20 | **-0.05** | -14.46 | 0.00 | -13.33 | 0.28 | -10.98 | 0.53 | -13.30 |
| Standard AE | 0.46 | -11.77 | 0.51 | -15.83 | 0.54 | -12.09 | 0.19 | -18.74 | -0.03 | -16.90 | 0.04 | -17.81 | 0.26 | -18.42 | 0.44 | -14.55 |
| Standard VAE | 0.47 | -10.61 | 0.63 | -12.50 | 0.25 | -8.63 | 0.10 | -14.66 | -0.05 | -15.72 | 0.02 | -16.04 | 0.18 | -15.53 | 0.07 | -14.43 |

In Appendix D, we provide a comprehensive ablation study showing the stability, scalability, and robustness of our MMD-based sculpting regularizer. As shown in Figure 9, while most baseline methods learn increasingly entangled representations as the latent space grows, our MMD regularizer maintains a consistently low LPS. Moreover, Figures 10 and 11 highlight the stability and robustness of the regularization as there is only a very small sensitivity in the model's LPS score and reconstruction NMSE when changing the training batch size from $64$ to $2048$, the regularization coefficient from $0.1$ to $5$, the number of bandwidth in the median heuristic from $1$ to $5$, and even the kernel type.

Lastly, Appendix E demonstrates the impact of changing the aggregate posterior's target prior on the alignment to semantic features of our unsupervised disentanglement method. Our experiments on

dsprites and XY(C/S) synthetic datasets demonstrate that the subsequent alignment of the representations can more than double with this simple change highlighting the fact that the choice of prior is, on its own, a powerful inductive bias for learning representations.

### 4.4 ALIGNMENT TO INTERPRETABLE FEATURES WITH THE PROGRAMMABLE PRIOR

While achieving (1) mutual independence is the primary objective of unsupervised methods, the ultimate goal of disentanglement is to (2) align these independent features with interpretable, real-world concepts. Our Programmable Prior Framework addresses this by allowing us to inject inductive bias via a semi-supervised mechanism described in Section 3 which balances three distinct training objectives with a simple non-parametric distribution matching regularizer.

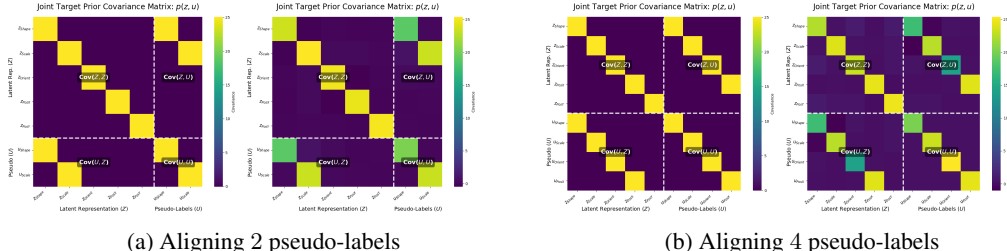

(a) Aligning 2 pseudo-labels        (b) Aligning 4 pseudo-labels

Figure 4: **Visualization of Alignment Mechanism in Programmable Prior Framework.** The figure shows the covariance matrix of the joint distribution $p(z, f(u))$ for the target (left) and the learned joint distribution (right) with 2 and 4 pseudo-labels respectively.

Formally, we construct a target prior where $z$ is sampled from a base distribution $p(z)$ (e.g., Gaussian or Uniform), and the semantic constraint is enforced by defining $f_\phi(u) = z_{0:k}$, where $k$ is the number of pseudo-labels to align. This joint pair $(z, f_\phi(u))$ acts both as a "glue" that aligns our model's latents $z$ with semantic features $f_\phi(u)$, and as the target prior that ensures the latent space is both mutually independent and correctly shaped (see visualization in Fig 4).

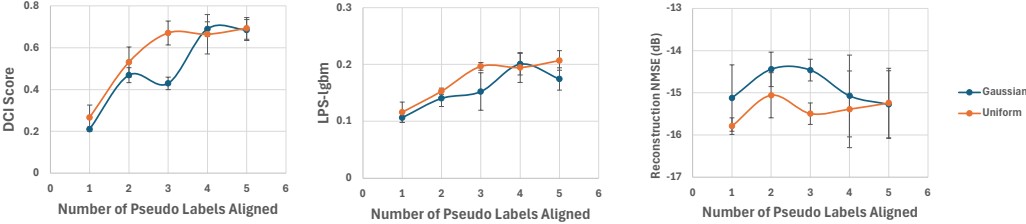

Figure 5: **Aligning the Latent Space to Semantic Features Using The Programmable Prior On dSprites.** The plots show the DCI Disentanglement Score (left), the Latent Predictability Score (LPS) using LGBM (middle), and the reconstruction NMSE in decibels (right) as the number of pseudo-labels used for alignment increases. The DCI scores were always evaluated with respect to all 5 semantic features (Shape, Scale, Orientation, X-Position, Y-Position).

We evaluate this mechanism on the dSprites dataset by progressively increasing the number of semantic factors (up to 5) for which we provide pseudo-labels. Figure 5 demonstrates the effectiveness of this approach. The DCI score rises sharply as we align more factors, reaching a peak when 3 out of 5 features are aligned for the uniform prior, and 4 out of 5 for the Gaussian prior. This confirms that our framework can successfully force the model to adopt a representation where specific latent dimensions correspond one-to-one with semantic factors, effectively addressing the NICA unidentifiability problem. Interestingly, as the model aligns with more semantic features, the LPS score increases (indicating *lower* statistical independence). This confirms a crucial insight from the literature: the "ground truth" factors in classic benchmarks like dSprites are often not statistically independent (e.g., size and position constraints). By forcing the model to align with these entangled

semantic concepts, we necessarily sacrifice pure statistical independence. This empirically validates our claim that Part 1 (Independence) and Part 2 (Alignment) are distinct and sometimes conflicting goals on the benchmark datasets. Throughout this process, the reconstruction error remains low and stable. This indicates that the model is able to find a representation that satisfies the alignment constraints without compromising its ability to compress and reconstruct the data.

### 4.5 QUANTIZER COMPATIBILITY WITH THE PROGRAMMABLE PRIOR

A critical advantage of our Programmable Prior Framework is its ability to sculpt the aggregate posterior into distributions inherently compatible with downstream tasks like compression. To demonstrate this, we trained our model with a quantization block in loop to quantize the encoder output. We utilize two commonly used quantizers: a uniform quantizer which considers a uniform distribution in the latent space ($\mathcal{U}[0,1]$) and a non-uniform quantizer, which considers a Gaussian distribution ($N(0,1)$). With our MMD-based framework, we constrain the latent space to have a latent distribution matched to the quantizer distribution. The downstream compression performance is measured by quantizing the learned latent codes to various bit-rates (2 to 8 bits per dimension) and computing the reconstruction NMSE in dB.

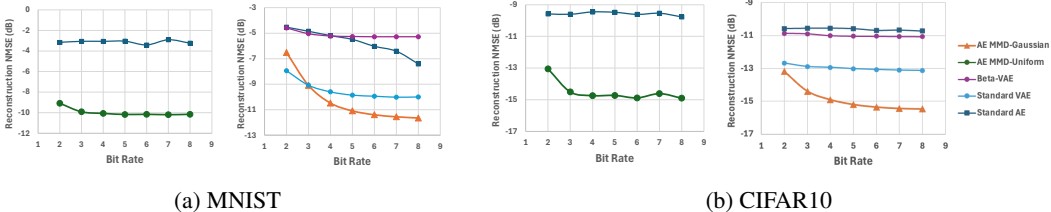

(a) MNIST                    (b) CIFAR10

Figure 6: **Compression with the Programmable Prior.** We compare the reconstruction NMSE (dB) of our MMD-regularized models against baselines with a uniform (left) or a Gaussian (right) quantizer.

As shown in Figure 6, our MMD-based models (blue lines) consistently outperforms the baseline by a wide margin ($\approx 2-7$ dB). This confirms our framework allows to explicitly program the prior to match the quantizer's expected distribution (whether Gaussian or Uniform), achieving high-fidelity compression "out of the box", without the need for complex post-training vector quantization or entropy coding schemes.

## 5 CONCLUSION

In this work, we challenged the foundational assumptions of the VAE-based paradigm for disentanglement, demonstrating that the standard KL-divergence is a flawed mechanism for enforcing statistical independence, the first step towards a disentangled representation. As a solution, we introduced the Programmable Prior Framework, a unified method built on the Maximum Mean Discrepancy that solves three distinct representation engineering goals: mutual independence, semantic alignment, and aggregate posterior distribution shaping. Our experiments confirm that this approach achieves state-of-the-art statistical independence on complex datasets without sacrificing reconstruction quality. Crucially, we showed that while independence is necessary, it is not sufficient for interpretability. By leveraging our framework's ability to enforce a programmable joint distribution, we demonstrated a novel semi-supervised mechanism to solve the alignment problem, effectively bridging the gap between statistical disentanglement and human-interpretable features. Ultimately, this work establishes the prior not as a fixed assumption, but as a precise engineering tool for sculpting latent spaces.

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

## APPENDIX

### LLM USE ACKNOWLEDGMENT

We acknowledge the use of Google's Gemini Pro 2.5 in the preparation of this manuscript. The model was utilized to aid in polishing grammar and clarity, assist with generating formatting for tables and figures, and serve as an assistant for the retrieval of prior work. The authors meticulously reviewed and edited all generated content and take full responsibility for the accuracy and originality of this work.

### TABLE OF CONTENTS

# A    Experiment and Implementation Details

## A.1    Experimental Design

### Datasets

Our method was evaluated on a diverse suite of benchmarks to assess its scalability and effectiveness.

- **Synthetic Datasets:** For controlled experiments with known ground-truth factors, we used **dSprites** (Matthey et al., 2017) and the **XY**, **XYC**, and **XYCS** environments (Cha & Thiyagalingam, 2023). These datasets feature simple shapes with explicit factors of variation, such as position (XY), color (C), and shape (S), and CelebA (Liu et al., 2015).

- **Real-World Datasets:** To test performance on complex, high-dimensional data, we used the standard image classification benchmarks **MNIST** (Deng, 2012), **CIFAR-10** (Krizhevsky et al., 2009), and **Tiny ImageNet** (Le & Yang, 2015).

### Baseline Models

We benchmarked our MMD-based approach against established methods for unsupervised disentanglement, including $\beta$-**VAE** (Higgins et al., 2017), $\beta$-**TCVAE** (Chen et al., 2018), **FactorVAE** (Kim & Mnih, 2018), and **DGAE** (Cha & Thiyagalingam, 2023). We also included a standard **Autoencoder (AE)** and a standard **VAE** as controls to establish a performance baseline without explicit disentanglement pressures.

### Training Details

All models were trained using the Adam optimizer with a learning rate of $5 \times 10^{-4}$ and a batch size of 512. Our MMD-based regularizer used a regularization coefficient of $\lambda = 0.3$. We additionally found that our framework was not very sensitive to variation in this hyperparameter as values between 0.1 and 2 resulted in comparable performance. The models are fully convolutional, featuring symmetric encoder-decoder architectures. Specific depths and latent dimensions for each dataset are detailed in Table 2. Finally, for the alignment mechanism, we employ a lightweight element-wise MLP for each pseudo-label with a single hidden layer of 64 units, structured as follows: Linear$(1, 64) \rightarrow$ BatchNorm $\rightarrow$ ReLU $\rightarrow$ Linear$(64, 1)$.

Table 2: Dataset-specific hyperparameters. The "Number of Layers" refers to the depth of the encoder and decoder individually (e.g., "6" indicates 6 layers for the encoder and 6 for the decoder).

| Dataset | Number of Layers | Latent Dimension ($d$) | Training Epochs |
|---|---|---|---|
| XY, XYC, XYCS | 6, 6, 6 | 2, 3, 4 | 500, 250, 125 |
| dSprites | 6 | 5 | 50 |
| CelebA | 6 | 32 | 75 |
| MNIST | 4 | 12 | 250 |
| CIFAR-10 | 4 | 64 | 250 |
| Tiny ImageNet | 6 | 64 | 75 |

## A.2    Evaluation Protocol and Metrics

To ensure robust and reproducible results, all reported scores are the mean and standard deviation over 5,10, or 15 independent runs with different random seeds.

### Metric Descriptions

**NMSE (dB / linear).** The Normalized Mean Squared Error measures reconstruction quality, comparing the output of the decoder to the original input image. We report it on both a linear scale and a decibel (dB) scale, which is logarithmic. For the dB scale, more negative values indicate better reconstruction.

**LPS (LGBM / MLP).** Our proposed Latent Predictability Score measures the mutual independence of the learned latent features. It is based on the average coefficient of determination ($R^2$) from a regression task that attempts to predict each latent dimension from all others. A score near zero indicates a high degree of independence. Therefore, a **lower LPS score is better**. We report two variants using LightGBM and a Multi-Layer Perceptron as the regressors.

**Covariance Ratio.** This metric quantifies the degree of *linear* feature independence by measuring how diagonal the latent covariance matrix is. It is calculated as the ratio of the sum of the diagonal elements to the sum of the absolute values of the off-diagonal elements. A **higher ratio indicates a more diagonal covariance matrix** and thus greater **linear** statistical independence.

**MIG (Mutual Information Gap).** This metric measures the degree to which each latent variable is informative about one, and only one, ground-truth factor. It is calculated as the un-normalized difference between the highest and second-highest mutual information over all factors Chen et al. (2018). A **higher MIG score is better**.

**DCI (Disentanglement, Completeness, Informativeness) (Eastwood & Williams, 2018).** This framework probes the latent space with a Random Forest regressor and reports three scores:

- **Disentanglement:** Measures if each latent dimension captures at most one ground-truth factor. A higher score is better.
- **Completeness:** Measures if each ground-truth factor is captured by a single latent dimension. A higher score is better.
- **Informativeness:** Measures the overall accuracy in predicting ground-truth factors from the latent representation. A higher score is better.

**SAP (Separated Attribute Predictability).** This metric measures the extent to which each latent dimension is predictive of only a single ground-truth factor. For each factor, it computes the difference in prediction error between a model trained on a single latent dimension and a model trained on all other dimensions. A **higher SAP score is better**.

### A.3 MMD REGULARIZER IMPLEMENTATION

Our MMD regularizer is designed to be robust and adaptive. The following provides specific implementation details.

**Kernel Configuration** We employ a mixture of Gaussian Radial Basis Function (RBF) kernels for all MMD calculations. The RBF kernel is defined as $k(z_i, z_j) = \exp(-\|z_i - z_j\|^2/(2\sigma^2))$, where $\sigma$ is the bandwidth. Using a sum of kernels with different bandwidths (generated from a base bandwidth with multipliers $[0.5, 1.0, 2.0]$) allows the MMD to capture distributional discrepancies across multiple scales, leading to a more robust distance metric.

**Adaptive Bandwidth Selection** Instead of fixing the kernel bandwidth $\sigma$, we use the **median heuristic** (Gretton et al., 2012). At each training step, the base bandwidth $\sigma_{\text{base}}$ is set to the median of all pairwise distances between the points in the combined set of latent samples $z \sim q_{\theta_1}(z)$ and prior samples $z' \sim p(z)$. This adaptive approach automatically scales the kernel to the current batch, eliminating the need for manual hyperparameter tuning.

**MMD Estimator** At each training step, we draw a batch of samples from the target prior $p(z)$ of the same size as the batch of encoded latent samples from the aggregate posterior $q_{\theta_1}(z)$. The MMD loss is then computed using the standard **biased, quadratic-time estimator** of the squared MMD. Despite its $O(n^2)$ complexity, this estimator is computationally efficient for typical batch sizes and is known to be effective and robust in practice.

# B COMPREHENSIVE EXPERIMENTAL RESULTS

This appendix provides the complete, unabridged quantitative results of our experiments for all models, and datasets. All reported scores are the mean and standard deviation computed over 5 independent runs with different random seeds.

Table 3: Summary for TINY IMAGENET

| Baseline | L=64 | | | | |
|---|---|---|---|---|---|
| | Covariance Ratio | LPS_LGBM | LPS_MLP | NMSE_dB | NMSE_linear |
| AE-MMD-Gaussian (Ours) | **54.386 ± 2.220** | **0.029 ± 0.002** | **0.001 ± 0.004** | **-11.704 ± 0.007** | **0.068 ± 0.000** |
| FactorVAE ($\gamma$=10.0) | 2.338 ± 0.286 | 0.437 ± 0.064 | 0.483 ± 0.065 | -10.754 ± 0.239 | 0.084 ± 0.005 |
| $\beta$-TCVAE ($\beta$=4.0) | 4.950 ± 0.264 | 0.794 ± 0.018 | 0.932 ± 0.015 | -9.798 ± 0.027 | 0.105 ± 0.001 |
| $\beta$-VAE ($\beta$=4) | 4.934 ± 0.104 | 0.726 ± 0.018 | 0.872 ± 0.035 | -9.866 ± 0.009 | 0.103 ± 0.000 |
| Standard AE | 7.699 ± 0.207 | 0.455 ± 0.006 | 0.460 ± 0.007 | -11.775 ± 0.005 | 0.066 ± 0.000 |
| Standard VAE | 2.409 ± 0.224 | 0.470 ± 0.073 | 0.518 ± 0.066 | -10.608 ± 0.361 | 0.087 ± 0.008 |

Table 4: Summary for CIFAR10

| Baseline | L=64 | | | | |
|---|---|---|---|---|---|
| | Covariance Ratio | LPS_LGBM | LPS_MLP | NMSE_dB | NMSE_linear |
| AE-MMD-Gaussian (Ours) | **65.917 ± 1.788** | **0.038 ± 0.004** | **0.025 ± 0.008** | **-15.615 ± 0.008** | **0.027 ± 0.000** |
| FactorVAE ($\gamma$=10.0) | 3.347 ± 0.075 | 0.628 ± 0.004 | 0.788 ± 0.006 | -12.491 ± 0.012 | 0.056 ± 0.000 |
| $\beta$-TCVAE ($\beta$=4.0) | 10.711 ± 0.350 | 0.890 ± 0.008 | 0.953 ± 0.006 | -10.684 ± 0.005 | 0.085 ± 0.000 |
| $\beta$-VAE ($\beta$=4) | 11.506 ± 0.238 | 0.846 ± 0.008 | 0.925 ± 0.006 | -10.492 ± 0.006 | 0.089 ± 0.000 |
| Standard AE | 6.929 ± 0.192 | 0.508 ± 0.005 | 0.513 ± 0.005 | -15.834 ± 0.007 | 0.026 ± 0.000 |
| Standard VAE | 3.405 ± 0.113 | 0.632 ± 0.009 | 0.793 ± 0.010 | -12.503 ± 0.009 | 0.056 ± 0.000 |

Table 5: Summary for MNIST

| Baseline | L=12 | | | | |
|---|---|---|---|---|---|
| | Covariance Ratio | LPS_LGBM | LPS_MLP | NMSE_dB | NMSE_linear |
| AE-MMD-Gaussian (Ours) | **46.635 ± 2.442** | **0.220 ± 0.007** | 0.333 ± 0.006 | **-11.922 ± 0.024** | **0.064 ± 0.000** |
| DGAE | 6.191 ± 0.368 | 0.632 ± 0.002 | 0.671 ± 0.005 | -11.478 ± 0.027 | 0.071 ± 0.000 |
| FactorVAE ($\gamma$=10.0) | 2.628 ± 0.094 | 0.271 ± 0.066 | **0.275 ± 0.098** | -8.621 ± 0.023 | 0.137 ± 0.001 |
| $\beta$-TCVAE ($\beta$=4.0) | 8.071 ± 2.238 | 0.838 ± 0.022 | 0.893 ± 0.015 | -5.967 ± 0.008 | 0.253 ± 0.000 |
| $\beta$-VAE ($\beta$=4) | 9.493 ± 1.000 | 0.898 ± 0.009 | 0.938 ± 0.007 | -4.606 ± 0.020 | 0.346 ± 0.002 |
| Standard AE | 8.353 ± 0.486 | 0.539 ± 0.007 | 0.587 ± 0.008 | -12.089 ± 0.020 | 0.062 ± 0.000 |
| Standard VAE | 2.656 ± 0.267 | 0.249 ± 0.053 | 0.242 ± 0.070 | -8.630 ± 0.011 | 0.137 ± 0.000 |

Table 6: Summary for CELEBA

| Baseline | L=32 | | | | | | | | |
|---|---|---|---|---|---|---|---|---|---|
| | NMSE_dB | NMSE_linear | completeness | disentanglement | feature_dependence | informativeness | lps_lgbm | lps_mlp | mig | sap |
| AE-MMD-Gaussian | -14.370 ± 0.012 | 0.037 ± 0.000 | 0.009 ± 0.000 | 0.008 ± 0.000 | 61.678 ± 3.671 | 0.868 ± 0.001 | 0.080 ± 0.003 | 0.227 ± 0.009 | 0.006 ± 0.002 | 0.000 ± 0.000 |
| DGAE | -14.404 ± 0.016 | 0.036 ± 0.000 | 0.014 ± 0.001 | 0.013 ± 0.001 | 5.742 ± 0.318 | 0.873 ± 0.001 | 0.481 ± 0.011 | 0.529 ± 0.010 | 0.011 ± 0.003 | 0.000 ± 0.000 |
| FactorVAE ($\gamma$=10.0) | -14.394 ± 0.150 | 0.036 ± 0.001 | 0.014 ± 0.001 | 0.013 ± 0.001 | 62.873 ± 2.077 | 0.871 ± 0.001 | 0.071 ± 0.007 | 0.106 ± 0.018 | 0.014 ± 0.003 | 0.000 ± 0.000 |
| Standard AE | -14.546 ± 0.012 | 0.035 ± 0.000 | 0.012 ± 0.001 | 0.012 ± 0.001 | 7.490 ± 0.169 | 0.871 ± 0.000 | 0.441 ± 0.004 | 0.479 ± 0.005 | 0.008 ± 0.001 | 0.000 ± 0.000 |
| VAE | -14.426 ± 0.153 | 0.036 ± 0.001 | 0.015 ± 0.002 | 0.014 ± 0.001 | 64.075 ± 3.956 | 0.871 ± 0.001 | 0.073 ± 0.018 | 0.100 ± 0.027 | 0.017 ± 0.004 | 0.000 ± 0.000 |
| $\beta$-TCVAE ($\beta$=4.0) | -13.097 ± 0.143 | 0.049 ± 0.002 | 0.013 ± 0.001 | 0.012 ± 0.001 | 62.598 ± 6.288 | 0.869 ± 0.000 | 0.594 ± 0.015 | 0.807 ± 0.020 | 0.015 ± 0.003 | 0.000 ± 0.000 |
| $\beta$-VAE ($\beta$=4) | -13.299 ± 0.019 | 0.047 ± 0.000 | 0.013 ± 0.001 | 0.012 ± 0.001 | 74.026 ± 2.330 | 0.870 ± 0.000 | 0.528 ± 0.021 | 0.747 ± 0.030 | 0.013 ± 0.002 | 0.000 ± 0.000 |

Table 7: Summary for DSPRITES

| Baseline | L=5 | | | | | | | | | |
|---|---|---|---|---|---|---|---|---|---|---|
| | Covariance Ratio | LPS_LGBM | LPS_MLP | MIG_mig_score | NMSE_dB | NMSE_linear | SAP_sap_score | completeness | disentanglement | informativeness |
| AE-MMD-Gaussian (Ours) | 102.265 ± 21.202 | 0.039 ± 0.002 | 0.033 ± 0.006 | 0.058 ± 0.024 | -17.888 ± 0.281 | 0.016 ± 0.001 | 0.000 ± 0.000 | 0.026 ± 0.018 | 0.025 ± 0.017 | 0.950 ± 0.002 |
| AE-MMD-hybrid-sampler (Ours) | 263.218 ± 87.577 | 0.105 ± 0.008 | 0.087 ± 0.012 | 0.247 ± 0.030 | -16.311 ± 0.261 | 0.023 ± 0.001 | 0.246 ± 0.027 | 0.200 ± 0.038 | 0.196 ± 0.038 | 0.957 ± 0.001 |
| DGAE | 4.065 ± 1.233 | 0.557 ± 0.112 | 0.573 ± 0.147 | 0.261 ± 0.198 | -14.614 ± 2.173 | 0.040 ± 0.026 | 0.000 ± 0.000 | 0.226 ± 0.104 | 0.219 ± 0.101 | 0.962 ± 0.004 |
| FactorVAE ($\gamma$=10.0) | 8.477 ± 1.004 | 0.082 ± 0.010 | 0.053 ± 0.017 | 0.226 ± 0.079 | -14.564 ± 0.110 | 0.035 ± 0.001 | 0.000 ± 0.000 | 0.081 ± 0.033 | 0.079 ± 0.032 | 0.964 ± 0.004 |
| $\beta$-TCVAE ($\beta$=4.0) | 6.469 ± 4.162 | 0.223 ± 0.165 | 0.180 ± 0.182 | 0.408 ± 0.215 | -8.831 ± 0.085 | 0.131 ± 0.003 | 0.000 ± 0.000 | 0.212 ± 0.119 | 0.195 ± 0.105 | 0.960 ± 0.005 |
| $\beta$-VAE ($\beta$=4) | 9.614 ± 5.771 | 0.525 ± 0.182 | 0.523 ± 0.214 | 0.529 ± 0.232 | -8.200 ± 0.057 | 0.000 ± 0.000 | 0.000 ± 0.000 | 0.236 ± 0.114 | 0.217 ± 0.100 | 0.956 ± 0.005 |
| Standard AE | 12.669 ± 2.560 | 0.187 ± 0.039 | 0.181 ± 0.042 | 0.078 ± 0.032 | -18.739 ± 0.290 | 0.013 ± 0.001 | 0.082 ± 0.058 | 0.041 ± 0.018 | 0.041 ± 0.018 | 0.957 ± 0.001 |
| Standard VAE | 5.528 ± 1.291 | 0.100 ± 0.016 | 0.047 ± 0.023 | 0.436 ± 0.365 | -14.661 ± 0.225 | 0.034 ± 0.002 | 0.000 ± 0.000 | 0.231 ± 0.194 | 0.225 ± 0.189 | 0.969 ± 0.011 |

Table 8: Summary for XY

| | | | | | L=2 | | | | | |
|---|---|---|---|---|---|---|---|---|---|---|
| Baseline | Covariance Ratio | LPS_LGBM | LPS_MLP | MIG_mig_score | NMSE_dB | NMSE_linear | SAP_sap_score | completeness | disentanglement | informativeness |
| AE-MMD-Gaussian (Ours) | 240.970 ± 243.891 | -0.053 ± 0.007 | -0.004 ± 0.003 | 0.751 ± 0.554 | -15.222 ± 0.403 | 0.030 ± 0.003 | 0.423 ± 0.316 | 0.199 ± 0.244 | 0.199 ± 0.243 | 1.000 ± 0.000 |
| AE-MMD-hybrid-sampler (Ours) | 380.827 ± 408.352 | -0.065 ± 0.020 | -0.003 ± 0.005 | 1.047 ± 0.414 | -15.825 ± 0.154 | 0.026 ± 0.001 | 0.713 ± 0.139 | 0.471 ± 0.230 | 0.469 ± 0.233 | 1.000 ± 0.000 |
| DGAE | 16.631 ± 19.417 | 0.248 ± 0.380 | 0.241 ± 0.385 | 1.580 ± 1.414 | -6.742 ± 2.522 | 0.262 ± 0.206 | 0.000 ± 0.000 | 0.521 ± 0.435 | 0.521 ± 0.435 | 0.999 ± 0.001 |
| FactorVAE (γ=10.0) | 2.022 ± 1.392 | -0.040 ± 0.011 | 0.015 ± 0.013 | 0.693 ± 0.545 | -15.364 ± 0.359 | 0.029 ± 0.002 | 0.537 ± 0.268 | 0.337 ± 0.255 | 0.336 ± 0.255 | 1.000 ± 0.000 |
| β-TCVAE (β=4.0) | 1.597 ± 0.551 | -0.042 ± 0.015 | 0.009 ± 0.006 | 1.149 ± 1.093 | -14.684 ± 0.278 | 0.034 ± 0.002 | 0.502 ± 0.320 | 0.372 ± 0.345 | 0.367 ± 0.344 | 1.000 ± 0.000 |
| β-VAE (β=4) | 2.424 ± 1.371 | -0.053 ± 0.013 | 0.016 ± 0.015 | 0.688 ± 0.643 | -14.457 ± 0.137 | 0.036 ± 0.001 | 0.522 ± 0.323 | 0.358 ± 0.339 | 0.358 ± 0.339 | 1.000 ± 0.000 |
| Standard AE | 503.878 ± 536.454 | -0.028 ± 0.037 | 0.024 ± 0.014 | 0.964 ± 0.837 | -16.904 ± 0.081 | 0.020 ± 0.000 | 0.470 ± 0.348 | 0.362 ± 0.369 | 0.362 ± 0.369 | 1.000 ± 0.000 |
| Standard VAE | 1.284 ± 0.134 | -0.055 ± 0.011 | 0.015 ± 0.007 | 0.195 ± 0.098 | -15.720 ± 0.273 | 0.027 ± 0.002 | 0.309 ± 0.057 | 0.103 ± 0.030 | 0.102 ± 0.030 | 1.000 ± 0.000 |

Table 9: Summary for XYC

| | | | | | L=3 | | | | | |
|---|---|---|---|---|---|---|---|---|---|---|
| Baseline | Covariance Ratio | LPS_LGBM | LPS_MLP | MIG_mig_score | NMSE_dB | NMSE_linear | SAP_sap_score | completeness | disentanglement | informativeness |
| AE-MMD-Gaussian (Ours) | 49.341 ± 12.168 | -0.014 ± 0.016 | 0.016 ± 0.007 | 0.256 ± 0.118 | -16.311 ± 0.245 | 0.023 ± 0.001 | 0.237 ± 0.077 | 0.115 ± 0.039 | 0.113 ± 0.038 | 0.999 ± 0.000 |
| AE-MMD-hybrid-sampler (Ours) | 324.644 ± 274.142 | 0.168 ± 0.030 | 0.117 ± 0.035 | 0.465 ± 0.336 | -15.962 ± 0.205 | 0.026 ± 0.240 | 0.233 ± 0.240 | 0.228 ± 0.224 | 0.175 ± 0.098 | 0.996 ± 0.005 |
| DGAE | 4.802 ± 2.926 | 0.498 ± 0.417 | 0.503 ± 0.414 | 0.340 ± 0.181 | -10.936 ± 3.421 | 0.110 ± 0.083 | 0.000 ± 0.000 | 0.184 ± 0.105 | 0.175 ± 0.098 | 0.996 ± 0.005 |
| FactorVAE (γ=10.0) | 1.581 ± 0.076 | -0.000 ± 0.006 | 0.032 ± 0.005 | 0.140 ± 0.059 | -15.919 ± 0.219 | 0.026 ± 0.001 | 0.183 ± 0.157 | 0.079 ± 0.069 | 0.076 ± 0.065 | 1.000 ± 0.000 |
| β-TCVAE (β=4.0) | 2.900 ± 0.258 | -0.010 ± 0.009 | 0.023 ± 0.007 | 0.188 ± 0.105 | -13.720 ± 0.129 | 0.042 ± 0.001 | 0.198 ± 0.120 | 0.093 ± 0.078 | 0.090 ± 0.075 | 1.000 ± 0.000 |
| β-VAE (β=4) | 1.861 ± 0.119 | 0.003 ± 0.010 | 0.027 ± 0.009 | 0.266 ± 0.149 | -13.333 ± 0.089 | 0.046 ± 0.001 | 0.278 ± 0.123 | 0.143 ± 0.107 | 0.138 ± 0.100 | 1.000 ± 0.000 |
| Standard AE | 9.459 ± 4.389 | 0.042 ± 0.019 | 0.072 ± 0.019 | 0.200 ± 0.082 | -17.811 ± 0.134 | 0.017 ± 0.001 | 0.227 ± 0.090 | 0.087 ± 0.052 | 0.085 ± 0.049 | 1.000 ± 0.000 |
| Standard VAE | 1.606 ± 0.125 | 0.016 ± 0.012 | 0.048 ± 0.008 | 0.176 ± 0.082 | -16.043 ± 0.147 | 0.025 ± 0.001 | 0.197 ± 0.056 | 0.086 ± 0.043 | 0.085 ± 0.043 | 0.999 ± 0.000 |

Table 10: Summary for XYCS

| | | | | | L=4 | | | | | |
|---|---|---|---|---|---|---|---|---|---|---|
| Baseline | Covariance Ratio | LPS_LGBM | LPS_MLP | MIG_mig_score | NMSE_dB | NMSE_linear | SAP_sap_score | completeness | disentanglement | informativeness |
| AE-MMD-Gaussian (Ours) | 42.117 ± 2.445 | 0.111 ± 0.011 | 0.106 ± 0.020 | 0.092 ± 0.036 | -16.757 ± 0.143 | 0.021 ± 0.001 | 0.110 ± 0.032 | 0.048 ± 0.016 | 0.048 ± 0.016 | 0.997 ± 0.000 |
| AE-MMD-hybrid-sampler (Ours) | 133.148 ± 52.617 | 0.203 ± 0.021 | 0.151 ± 0.025 | 0.235 ± 0.070 | -16.101 ± 0.283 | 0.025 ± 0.002 | 0.278 ± 0.060 | 0.149 ± 0.051 | 0.146 ± 0.050 | 0.998 ± 0.000 |
| DGAE | 7.087 ± 3.365 | 0.586 ± 0.235 | 0.583 ± 0.233 | 0.275 ± 0.159 | -15.023 ± 0.701 | 0.032 ± 0.005 | 0.000 ± 0.000 | 0.190 ± 0.076 | 0.187 ± 0.074 | 0.997 ± 0.001 |
| FactorVAE (γ=10.0) | 1.609 ± 0.127 | 0.167 ± 0.020 | 0.165 ± 0.032 | 0.176 ± 0.114 | -15.517 ± 0.366 | 0.028 ± 0.002 | 0.192 ± 0.090 | 0.149 ± 0.098 | 0.148 ± 0.097 | 0.997 ± 0.001 |
| β-TCVAE (β=4.0) | 1.770 ± 0.411 | 0.171 ± 0.024 | 0.148 ± 0.028 | 0.277 ± 0.115 | -12.402 ± 0.257 | 0.058 ± 0.003 | 0.227 ± 0.109 | 0.246 ± 0.108 | 0.245 ± 0.107 | 0.999 ± 0.000 |
| β-VAE (β=4) | 1.224 ± 0.050 | 0.276 ± 0.063 | 0.258 ± 0.073 | 0.396 ± 0.065 | -10.983 ± 0.672 | 0.081 ± 0.012 | 0.238 ± 0.053 | 0.264 ± 0.031 | 0.262 ± 0.031 | 0.988 ± 0.006 |
| Standard AE | 12.609 ± 4.281 | 0.265 ± 0.028 | 0.284 ± 0.031 | 0.099 ± 0.052 | -18.422 ± 0.279 | 0.014 ± 0.001 | 0.130 ± 0.081 | 0.069 ± 0.029 | 0.068 ± 0.028 | 0.998 ± 0.000 |
| Standard VAE | 1.630 ± 0.175 | 0.179 ± 0.042 | 0.165 ± 0.051 | 0.163 ± 0.077 | -15.531 ± 0.432 | 0.028 ± 0.003 | 0.184 ± 0.079 | 0.106 ± 0.062 | 0.105 ± 0.061 | 0.996 ± 0.001 |

864
865
866
867
868
869
870
871

## C  METRIC ANALYSIS

This section provides a deeper analysis of the properties of common disentanglement metrics, justifying the introduction and use of our Latent Predictability Score (LPS). While alignment-based metrics like DCI and MIG are popular, our experiments reveal that they can exhibit paradoxical behavior and high instability, potentially leading to misleading conclusions about the quality of a learned representation.

872
873

### C.1  THE ALIGNMENT-INDEPENDENCE PARADOX

874
875
876
877
878
879
880
881
882
883
884
885
886
887
888
889
890

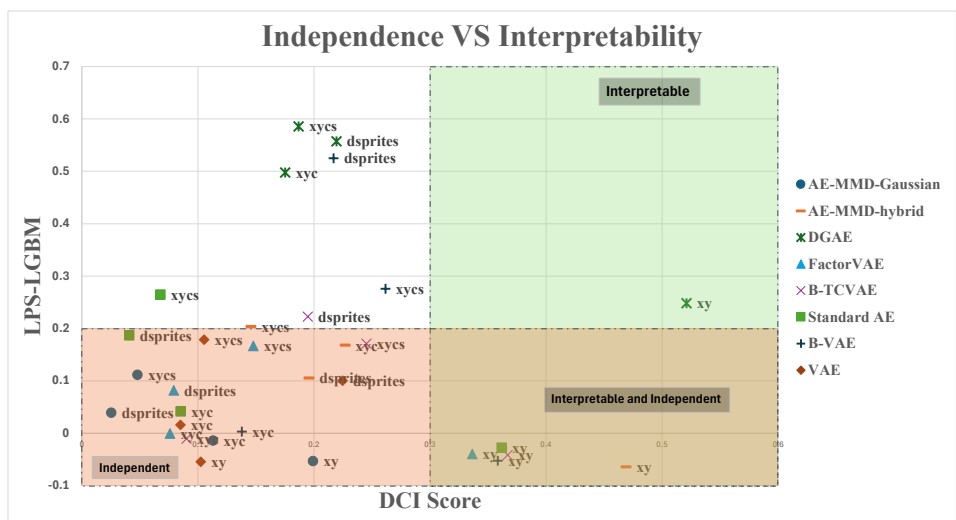

891
892
893
894
895
896

Figure 7: **The Paradox of Independence vs. Alignment.** Each point represents a model. The ideal model would be in the bottom-right (high alignment, high independence). We observe that models like DGAE occupy the unfavorable top-left quadrant, achieving a high DCI score only by sacrificing independence. Our method (with a Gaussian prior) occupies the bottom-left, prioritizing true independence over potentially flawed alignment metrics.

897
898
899
900

A core assumption is that a higher alignment score (like DCI) corresponds to a better, more disentangled representation. Our findings challenge this notion, revealing a troubling paradox: optimizing for alignment can actively work against the goal of learning statistically independent factors.

901
902
903
904
905
906
907
908
909
910
911

This effect is visualized in the scatter plot in Figure 7. The ideal model would occupy the bottom-right quadrant, signifying both high alignment (high DCI) and high independence (low LPS). However, we observe that baseline models frequently land in the bottom-left or top-left "paradox zone." A clear example is DGAE on the XY dataset, which achieves the highest DCI score (0.520) but does so with a highly entangled representation (LPS of 0.25) and poor reconstruction. This suggests that DCI may be rewarding the model for learning spurious correlations, an effect conjectured in past work which noted that many benchmark datasets contain significant correlations between their ground-truth factors (Locatello et al., 2019). For instance, in dSprites and XYCS, an object's size and position are co-dependent (a large shape cannot be centered near an edge). A model that correctly learns this co-dependence may score well on DCI but has fundamentally failed to learn truly independent factors. This highlights a critical flaw in relying solely on alignment-based metrics as a proxy for disentanglement.

912
913
914

### C.2  METRIC INSTABILITY ACROSS RUNS

915
916
917

Beyond the paradox, we find that standard alignment metrics are highly unstable, exhibiting extreme variance across different random seeds. This is quantified and visualized in Figure 8. As shown in Figure 8(a), the standard deviation of the DCI score for baseline methods is often as large as, or even larger than, the mean itself. This instability, a known symptom of the NICA identifiability problem

Table 11: Comparison of Disentanglement (DCI) and Latent Predictability Score (LPS) across models and synthetic datasets. For Disentanglement, higher is better (↑), and for LPS, lower is better (↓).

| | dsprites | | xy | | xyc | | xycs | |
|---|---|---|---|---|---|---|---|---|
| **Baseline** | Disent. (↑) | LPS (↓) | Disent. (↑) | LPS (↓) | Disent. (↑) | LPS (↓) | Disent. (↑) | LPS (↓) |
| **AE-MMD-Gaussian** | 0.025 | **0.039** | 0.199 | -0.053 | 0.113 | **-0.014** | 0.048 | **0.111** |
| **AE-MMD-hybrid** | 0.196 | 0.105 | 0.469 | **-0.065** | **0.227** | 0.168 | 0.146 | 0.203 |
| DGAE | **0.219** | 0.557 | **0.521** | 0.248 | 0.175 | 0.498 | 0.187 | 0.596 |
| FactorVAE | 0.079 | 0.082 | 0.336 | -0.040 | 0.076 | -0.000 | 0.148 | 0.187 |
| $\beta$-TCVAE | 0.195 | 0.222 | 0.368 | -0.042 | 0.090 | -0.010 | 0.245 | 0.171 |
| $\beta$-VAE | 0.217 | 0.525 | 0.358 | -0.053 | 0.138 | 0.003 | **0.262** | 0.276 |
| Standard AE | 0.041 | 0.187 | 0.362 | -0.028 | 0.085 | 0.042 | 0.068 | 0.265 |
| Standard VAE | 0.225 | 0.100 | 0.102 | -0.055 | 0.085 | 0.016 | 0.105 | 0.180 |

(Locatello et al., 2019), makes scores from single runs untrustworthy and complicates reliable model comparison.

In stark contrast, Figure 8(b) shows that our LPS metric is highly stable, with consistently small error bars across all models and datasets. This stability stems from the fact that LPS measures an intrinsic, unsupervised property of the representation itself (mutual independence) rather than its alignment to a specific, and potentially arbitrary, set of ground-truth labels.

## C.3 LPS as a Stable, Intrinsic Measure

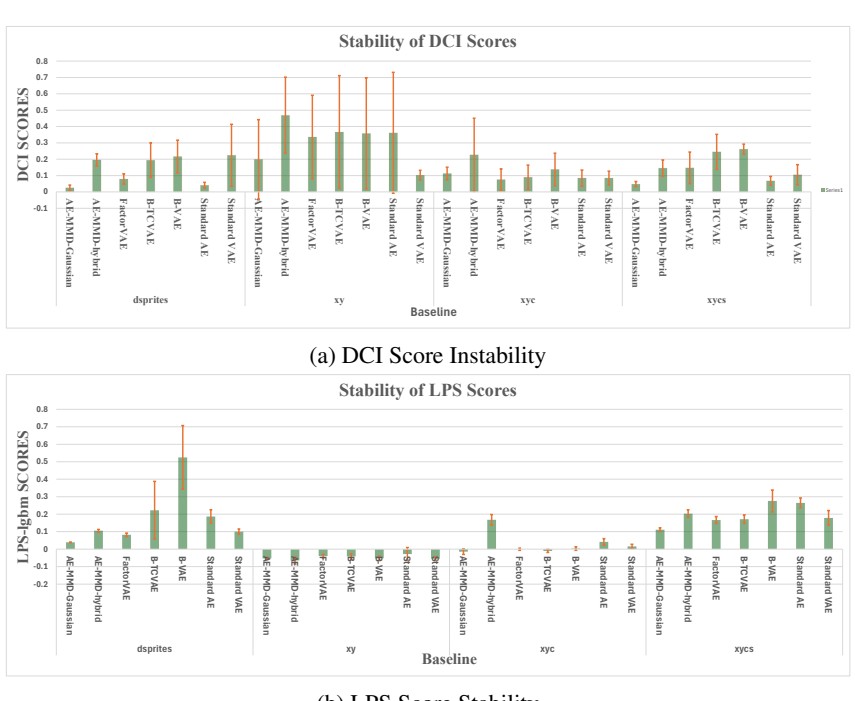

(a) DCI Score Instability

(b) LPS Score Stability

Figure 8: **Comparison of metric stability across 5 random seeds.** The error bars represent one standard deviation. **(a)** The DCI scores exhibit extremely high variance, with the standard deviation often being as large as the mean itself, highlighting the metric's unreliability. **(b)** In stark contrast, our proposed LPS metric shows minimal variance, demonstrating its stability and reliability as an intrinsic measure of representation quality.

The paradoxical behavior and instability of alignment-based metrics underscore the need for a more fundamental and reliable measure of representation quality. We argue that the LPS fills this critical gap. By focusing on statistical independence, LPS provides a stable, intrinsic, and fully unsupervised signal. It directly answers the question, "Are the learned features statistically independent?"

without being confounded by the potential issues of the ground-truth labels or the inherent instability of the alignment problem. While alignment is a desirable downstream goal, achieving a verifiably independent representation is a more foundational first step, and LPS provides the tool to measure it robustly.

# D SCALABILITY, STABILITY, AND ROBUSTNESS OF THE MMD-BASED SCULPTING REGULARIZER

In the main text, we demonstrated that the Programmable Prior Framework achieves state-of-the-art disentanglement. In this section, we provide a comprehensive ablation study to verify the stability of the MMD Sculpting regularizer. Since MMD is a sample-based, non-parametric estimator, it is crucial to examine its behavior with respect to three key implementation factors:

1. **Model Capacity:** Does the latent space become entangled as the dimension increases?

2. **Batch Size:** Does the MMD estimator require prohibitively large batches to converge?

3. **Kernel Choice:** Is the method sensitive to the specific choice of kernel function or bandwidth hyperparameters?

The following experiments were conducted on the CIFAR-10 dataset using the LPS-lgbm metric (unless otherwise stated) to quantify mutual independence, alongside the reconstruction NMSE (dB) to verify that any gains in disentanglement do not come at the cost of model performance.

ROBUSTNESS TO LATENT SPACE DIMENSIONALITY

We first investigate the "capacity problem" common in VAEs, where increasing the latent dimension often leads to posterior collapse or increased entanglement. We trained models with latent dimensions ranging from $d = 32$ to $d = 96$.

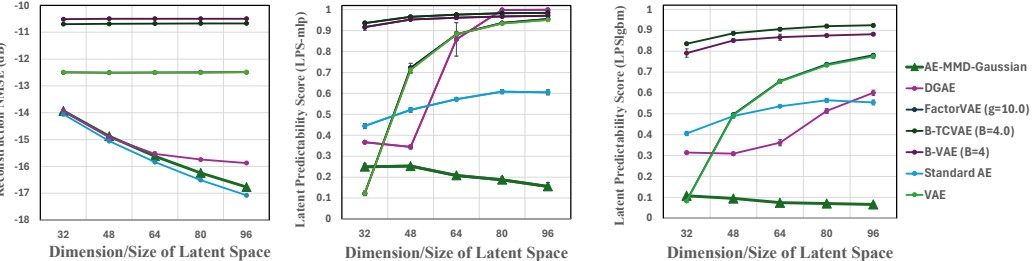

Figure 9: **Latent Independence vs. Latent Space Dimension on CIFAR-10.** The plots show the reconstruction NMSE in decibels (left), and the Latent Predictability Score (LPS) using MLP (middle) and LGBM (right) regression as the size of the latent bottleneck increases. Lower scores are better for all metrics. Our method (AE-MMD-Gauss, green line) maintains a high degree of independence across all dimensions, while most baselines learn increasingly entangled representations. Moreover, we achieve this stability at no significant cost to reconstruction error, which is surpassed only by the unregularized standard AE model.

As shown in Figure 9, our MMD-based framework (green line with triangles) exhibits unique robustness:

- **Independence (Middle/Right Plots):** The LPS scores for our model remain consistently low ($< 0.25$) across all dimensions. In stark contrast, baseline methods like $\beta$-TCVAE and FactorVAE show clear degradation, with LPS scores rising towards $1.0$ (indicating complete predictability/entanglement) as the latent space grows.

- **Reconstruction (Left Plot):** Crucially, our method does not sacrifice reconstruction for this independence. The NMSE continues to improve (becoming more negative) as the dimension increases, tracking the performance of an unregularized Autoencoder (blue line). This contrasts with the standard VAE (light green circle), which plateaus at a higher error rate regardless of capacity.

SENSITIVITY TO BATCH SIZE AND REGULARIZATION STRENGTH

The MMD is a sample-based statistic, meaning its variance depends on the batch size $N$. We evaluated the method's performance across batch sizes ranging from 64 to 2048 and regularization coefficients from 0.1 to 5.

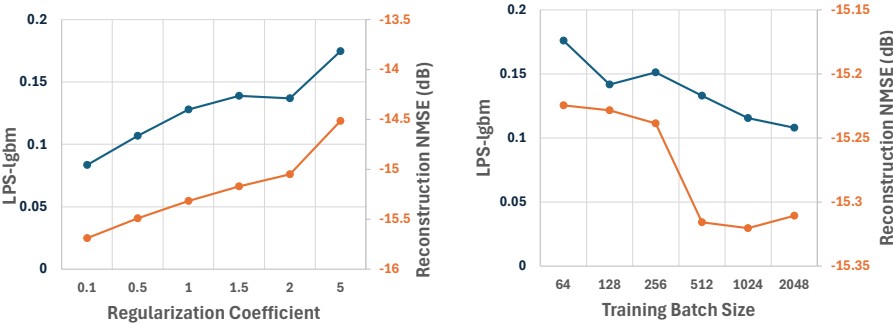

Figure 10: **Ablation Study on CIFAR-10: Regularization Coefficient and Batch Size.** We report the LPS-lgbm score and the test reconstruction NMSE (dB) against regularization coefficients ranging from $0.1$ to $5$ (left), and training batch sizes ranging from $64$ to $2048$ (right).

Figure 10 reveals two key insights:

- **Independence (Blue Line):** There is a slight improvement in disentanglement (LPS drops from $\approx 0.17$ to $\approx 0.11$) as the batch size increases from 64 to 2048, or as the regularization coefficient decreases from 5 down to 0.1. This indicates some sensitivity, which is to be expected, but primarily highlights the robustness of the MMD regularizer to sample complexity per training iteration.

- **Reconstruction (Orange Line):** The reconstruction quality is remarkably stable, varying by less than $0.1$ dB across the entire range of batch sizes and by just over $1$ dB across the range of regularization coefficients. This suggests that even with smaller batches, the regularization does not overpower the reconstruction objective, indicating that the MMD objective to sculpt the latent space is not misaligned with the goal of reconstruction.

ROBUSTNESS TO KERNEL CHOICE

A common criticism of MMD is the need to tune kernel bandwidths. To address this, we utilize a mixture of kernels with the median heuristic (as detailed in Appendix A.3). Figure 11 presents a scatter plot of performance for various kernel configurations.

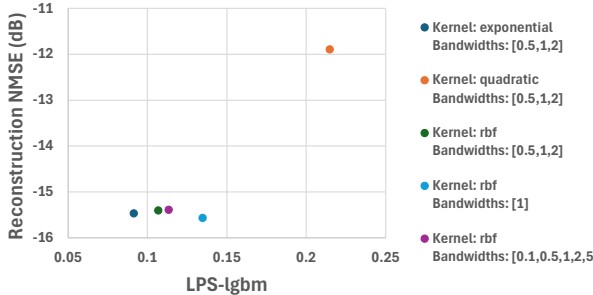

Figure 11: **Ablation Study of Kernel Type on CIFAR-10.** We report a scatter plot of the LPS-lgbm score versus the test reconstruction NMSE (dB) for exponential, quadratic, and Gaussian (RBF) kernel types. We also compare configurations with different numbers of bandwidths for kernel averaging according to the median heuristic.

The results demonstrate high stability:

- **Optimal Cluster:** The majority of configurations—including the Quadratic kernel and RBF kernels with widely varying bandwidth sets (e.g., single bandwidth vs. mixture of 5)—cluster tightly in the optimal bottom-left region (Low LPS $\approx 0.1$, Low NMSE $\approx -15.5$ dB).

- **Insensitivity to Bandwidth:** There is a small but statistically significant performance difference between using a single bandwidth (teal dot) and a mixture of 3 or even 5 bandwidths (purple dot). However, the difference between using 3 bandwidths versus 5 is negligible, validating the robustness of the median heuristic.

- **Outliers:** The Quadratic kernel (orange dot) performs noticeably worse in both reconstruction and independence at the same batch size and regularization coefficient, confirming that MMD is sensitive to the choice of kernel type (RBF/Exponential vs. Quadratic). However, we note that the quadratic kernel is rarely used for MMD applications compared to exponential or RBF kernels.

# E  THE POWER OF PROGRAMMABLE PRIORS: EXTENDED RESULTS

This section provides a deeper dive into the concept of "programmable priors," showcasing (1) the flexibility of our MMD framework and (2) how just the choice of the target prior can significantly improve the unsupervised alignment of learned representations with desirable semantic factors. We present results where the latent space was sculpted to match a custom set of per-dimension priors, reflecting a better inductive bias for each dataset. The following visual and quantitative data will demonstrate that this prior engineering significantly improves ground-truth alignment metrics over the baseline `AE-MMD-Gaussian` model.

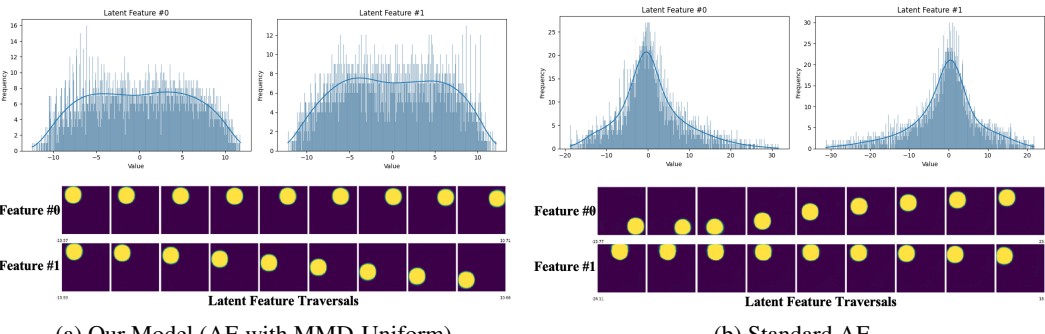

(a) Our Model (AE with MMD-Uniform)                    (b) Standard AE

Figure 12: **Qualitative Comparison on the XY Dataset.** Each column compares the learned latent marginals (top) with the corresponding latent traversals (bottom). **(a)** Our MMD-trained model successfully learns the true uniform distribution, which results in clean, disentangled traversals. **(b)** In contrast, the standard AE defaults to a uni-modal distribution, leading to entangled traversals.

Figure 12 provides a clear demonstration on the XY dataset (we provide additional results for the other synthetic datasets in the later subsections), where the ground-truth factor (position) follows a uniform distribution. When we program our prior to be uniform, our model successfully sculpts the latent marginals to match this target geometry. This is in stark contrast to a standard autoencoder, which defaults to a uni-modal representation that fails to capture the data structure with the vertical and horizontal x and y axis. The aligned latent space is evident in the latent traversals: our model's traversals are clean and disentangled, while the standard AE's are entangled and less interpretable.

Table 12: Results for our MMD-based Autoencoder on the XY dataset with different engineered prior distributions. We find selecting a Uniform target prior significantly improve alignment metrics.

| Prior | NMSE (dB) | LPS-lgbm | LPS-mlp | DCI-RF | MIG |
|---|---|---|---|---|---|
| AE-Gaussian | -15.2 | -0.053 | **-0.004** | 0.199 | 0.751 |
| AE-Uniform | **-15.8** | **-0.064** | -0.003 | **0.469** | **1.047** |
| AE-GMM (2 component) | -14.9 | -0.051 | 0.000 | 0.251 | 0.863 |

The quantitative results in Table 12 confirm that this improved structure directly translates to better alignment. On the XY dataset, using a standard factorized Gaussian prior achieves excellent statistical independence (LPS of -0.05) but yields a modest alignment score (DCI of 0.199). However, by simply switching to a Uniform prior that mirrors the true generative process, we dramatically improve the DCI and MIG scores, achieving near-perfect alignment without sacrificing independence. This result underscores a key contribution of our work: the prior is a critical tool for representation engineering, and our framework is unique in its ability to program this bias to achieve interpretable alignment.

## E.1  DSPRITES: ENGINEERING A MIXED-DISTRIBUTION PRIOR

For the 5-dimensional latent space of dSprites, we designed a hybrid prior to test the model's ability to learn a mix of simple and multi-modal distributions. The visual results in Figure 13 confirm that the model's learned marginals consistently lead to improved recovery of desired latent features.

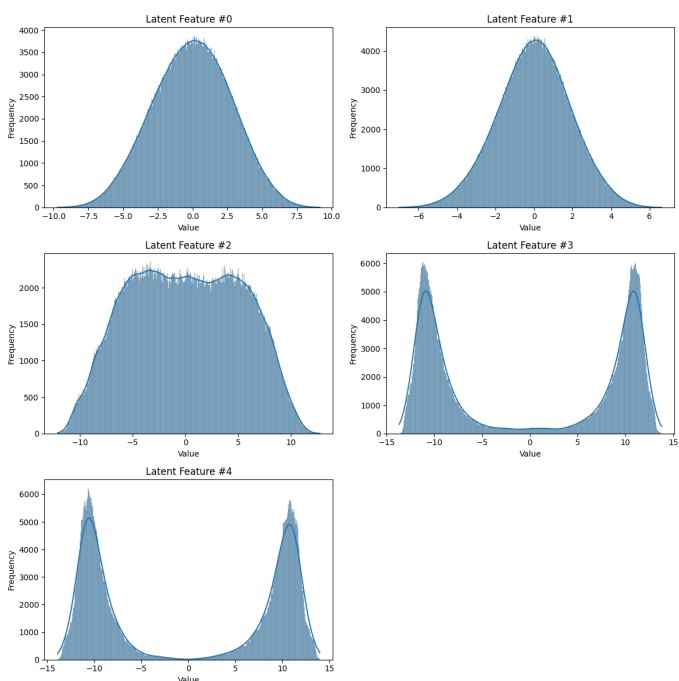

Figure 13: Learned marginal distributions for the `AE-MMD-hybrid-sampler` on dSprites. The histograms provide direct visual proof that the model successfully learned to match the specified target distributions for each latent dimension.

**Analysis.** The figure shows a direct correspondence between our specified priors and the learned representations:

- **Dimension 1 (Uniform):** We specified a `Uniform(-5, 5)` prior. The model chose to ignore this instruction during training and instead adopted a Gaussian distribution with the same mean and variance.

- **Dimension 2 (Gaussian):** The target was a wide `Gaussian(0, 2)`, and the model learned the corresponding bell curve.

- **Dimensions 3 (Bi-modal GMMs):** For this dimensions, we used `auto-gmm` priors with two modes centered at -5 and 5, and with standard deviation equal to 5. The histograms clearly show that the model learned to partition the data into a more uniform distribution. We have found that specifying such a GMM structure would generally help the model adhere to a more uniform representation. while specifying the uniform distribution directly as in Dimension 1 would sometimes be 'ignored'.

- **Dimensions 4-5 (Bi-modal GMMs):** For the remaining dimensions, we used `auto-gmm` priors with two modes located at -10 and 10 respectively with standard deviation of 1 (using a small standard deviation of 1 is used to ensure the model adopts a bi-modal representation and not a uniform distribution as observed in Dimension 3) . The histograms clearly show that the model learned to partition the data into these distinct modes for each respective dimension.

**Quantitative Results.** As shown in Table 13, this engineered prior leads to a dramatic improvement in alignment metrics. The `AE-MMD-hybrid-sampler` achieves a MIG score of 0.247, an over **4x improvement** compared to the Gaussian prior's score of 0.058. Similar substantial gains are seen in the SAP score (0.246 vs. 0.000) and the DCI disentanglement score (0.196 vs. 0.025), demonstrating that providing a better inductive bias directly translates to a more interpretable and aligned representation.

Table 13: Summary for DSPRITES

| | L=5 | | | | | | | | | |
| Baseline | Covariance Ratio | LPS_LGBM | LPS_MLP | MIG_mig_score | NMSE_dB | NMSE_linear | SAP_sap_score | completeness | disentanglement | informativeness |
| --- | --- | --- | --- | --- | --- | --- | --- | --- | --- | --- |
| AE-MMD-Gaussian | 102.265 ± 21.202 | 0.039 ± 0.002 | 0.033 ± 0.006 | 0.058 ± 0.024 | -17.888 ± 0.281 | 0.016 ± 0.001 | 0.000 ± 0.000 | 0.026 ± 0.018 | 0.025 ± 0.017 | 0.950 ± 0.002 |
| AE-MMD-hybrid-sampler | 263.218 ± 87.577 | 0.105 ± 0.008 | 0.087 ± 0.012 | 0.247 ± 0.030 | -16.311 ± 0.261 | 0.023 ± 0.001 | 0.246 ± 0.027 | 0.200 ± 0.038 | 0.196 ± 0.038 | 0.957 ± 0.001 |

## E.2 XY: ENGINEERING A UNIFORM PRIOR

We provide the full table of results for the XY experiment.

Table 14: Summary for XY

| | L=2 | | | | | | | | | |
| Baseline | Covariance Ratio | LPS_LGBM | LPS_MLP | MIG_mig_score | NMSE_dB | NMSE_linear | SAP_sap_score | completeness | disentanglement | informativeness |
| --- | --- | --- | --- | --- | --- | --- | --- | --- | --- | --- |
| AE-MMD-Gaussian | 240.970 ± 243.891 | -0.053 ± 0.007 | -0.004 ± 0.003 | 0.751 ± 0.554 | -15.222 ± 0.403 | 0.030 ± 0.003 | 0.423 ± 0.316 | 0.199 ± 0.244 | 0.199 ± 0.243 | 1.000 ± 0.000 |
| AE-MMD-hybrid-sampler | 380.827 ± 408.352 | -0.065 ± 0.020 | -0.003 ± 0.005 | 1.047 ± 0.414 | -15.825 ± 0.154 | 0.026 ± 0.001 | 0.713 ± 0.139 | 0.471 ± 0.230 | 0.469 ± 0.233 | 1.000 ± 0.000 |

## E.3 XYC: ENGINEERING A GMM-HEAVY PRIOR

On the XYC dataset, we used a prior dominated by bi-modal distributions to encourage the model to discover clustered structures. Figure 14 visualizes the successful enforcement of this prior.

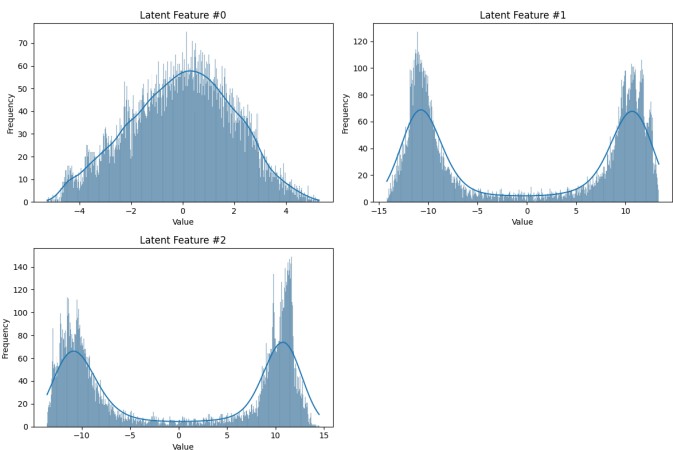

Figure 14: Learned marginal distributions for the `AE-MMD-hybrid-sampler` on XYC.

**Analysis.** The learned marginals again match the specified inductive bias:

- **Dimension 1 (Gaussian):** The model correctly learns a `Gaussian(0, 2)` distribution.
- **Dimensions 2 & 3 (Bi-modal GMMs):** We specified two identical `auto-gmm` priors with modes at -10 and 10. The resulting histograms clearly show two distinct clusters centered at these values, demonstrating the model's ability to learn multi-modal representations.

**Quantitative Results.** The quantitative results in Table 15 confirm the benefit of this custom prior. The hybrid-sampler model nearly doubles the MIG score (0.465 vs. 0.256) and the DCI disentanglement score (0.227 vs. 0.113) compared to the model trained with a simple Gaussian prior.

Table 15: Summary for XYC

| | L=3 | | | | | | | | | |
| Baseline | Covariance Ratio | LPS_LGBM | LPS_MLP | MIG_mig_score | NMSE_dB | NMSE_linear | SAP_sap_score | completeness | disentanglement | informativeness |
| --- | --- | --- | --- | --- | --- | --- | --- | --- | --- | --- |
| AE-MMD-Gaussian | 49.341 ± 12.168 | -0.014 ± 0.016 | 0.016 ± 0.007 | 0.256 ± 0.118 | -16.311 ± 0.245 | 0.023 ± 0.001 | 0.237 ± 0.077 | 0.115 ± 0.039 | 0.113 ± 0.038 | 0.999 ± 0.000 |
| AE-MMD-hybrid-sampler | 324.644 ± 274.142 | 0.168 ± 0.030 | 0.117 ± 0.035 | 0.465 ± 0.336 | -15.962 ± 0.205 | 0.025 ± 0.001 | 0.266 ± 0.240 | 0.228 ± 0.224 | 0.227 ± 0.224 | 1.000 ± 0.000 |

## E.4 XYCS: ENGINEERING A MIXED-COMPONENT PRIOR

For the 4-dimensional XYCS dataset, we again specified a mixed set of priors. The results in Figure 15 show that the model faithfully learns this complex target geometry.

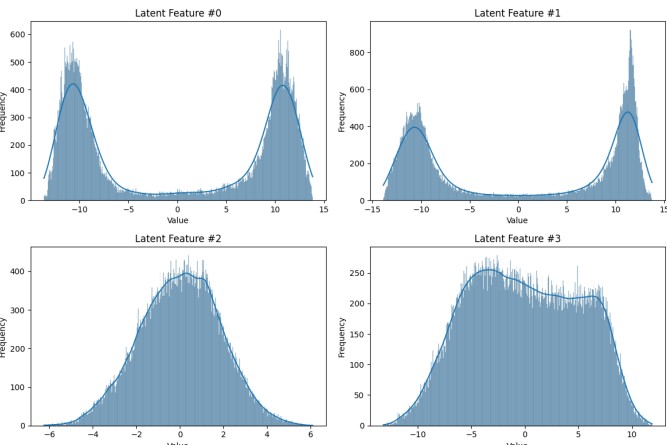

Figure 15: Learned marginal distributions for the `AE-MMD-hybrid-sampler` on XYCS.

**Analysis.** The learned distributions follow the specified configuration: two bi-modal GMMs with modes at $\pm 10$, a standard Gaussian, and a narrower bi-modal GMM with modes at $\pm 5$. Each histogram in the figure visually confirms that the intended structure was successfully imposed on the corresponding latent dimension.

**Quantitative Results.** As shown in Table 16, injecting this more informed inductive bias again leads to superior alignment. The hybrid-sampler model achieves a MIG score of 0.235, more than doubling the 0.092 score from the Gaussian model. Likewise, the SAP and DCI scores show significant improvements of over 2.5x and 3x, respectively.

Table 16: Summary for XYCS

| | L=4 | | | | | | | | | |
|---|---|---|---|---|---|---|---|---|---|---|
| Baseline | Covariance Ratio | LPS_LGBM | LPS_MLP | MIG_mig_score | NMSE_dB | NMSE_linear | SAP_sap_score | completeness | disentanglement | informativeness |
| AE-MMD-Gaussian | 42.117 ± 2.445 | 0.111 ± 0.011 | 0.106 ± 0.020 | 0.092 ± 0.036 | -16.757 ± 0.143 | 0.021 ± 0.001 | 0.110 ± 0.032 | 0.048 ± 0.016 | 0.048 ± 0.016 | 0.997 ± 0.000 |
| AE-MMD-hybrid-sampler | 133.148 ± 52.617 | 0.203 ± 0.021 | 0.151 ± 0.025 | 0.235 ± 0.070 | -16.101 ± 0.283 | 0.025 ± 0.002 | 0.278 ± 0.060 | 0.149 ± 0.051 | 0.146 ± 0.050 | 0.998 ± 0.000 |

## SUMMARY

The results in this section highlight a key takeaway: while a factorized Gaussian prior is highly effective at achieving statistical independence, it is not always the optimal choice for interpretability and alignment with ground-truth factors. The true power of the MMD framework is its flexibility, which allows practitioners to inject domain knowledge by engineering specific latent geometries. This capability leads to representations that are not only independent (or have a desired dependence structure) but are also more meaningfully aligned with the true factors of variation in the data.

However, our experiments with programmable priors also revealed a nuance in the MMD framework's behavior. While the regularizer consistently learned unimodal, Gaussian-like distributions with high fidelity, enforcing more specific geometries, such as uniform or multi-modal distributions, could sometimes present challenges. In these cases, the model occasionally defaulted to a unimodal distribution that correctly matched the target's mean and variance but not its overall shape.

We hypothesize this is primarily due to the inductive bias of the RBF kernel. As the RBF kernel is itself a Gaussian function, it is naturally more adept at matching distributions defined by their low-order moments (Tolstikhin et al., 2017). A secondary factor may be architectural constraints; the purely convolutional nature of the encoder might limit its capacity to shape the latent space into more complex geometries. We believe that future work could explore alternative kernels or

incorporate fully connected layers to enhance the encoder's ability to model a wider variety of prior distributions.

# F ADDITIONAL EXPERIMENTS

## F.1 THE PROGRAMMABLE PRIOR FOR REPRESENTATIONAL KNOWLEDGE TRANSFER

In the main text, we discussed our framework's ability to "copy" the latent distribution of a target model. Here, we provide this experiment's results on matching the aggregate prior.

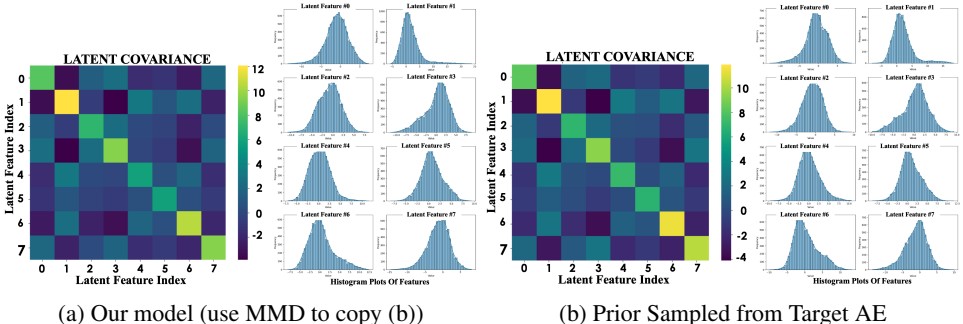

(a) Our model (use MMD to copy (b))     (b) Prior Sampled from Target AE

Figure 16: **Visualizing the Latent Space Copying experiment on MNIST.** (b) First, a standard Autoencoder was trained, and its complex, entangled latent distribution was saved to serve as an empirical prior. (a) Then, a new "student" model was trained using our MMD-based sculpting regularizer, tasked with replicating this non-analytic prior. The figure displays the covariance matrix between the latent space features over the whole dataset and the histogram plots of the marginal distribution of the latent space features over the whole dataset. The near-identical covariance matrices and marginal distributions visually confirm our method's ability to enforce a complex, co-dependent latent geometry that would be impossible to specify analytically.

We see that the model was able to match the target prior's covariance structure exactly while also closely matching the marginal histogram plots indicating that the Programmable Prior Framework is a powerful for being able to capture complex internal dependence structures and non analytic target distributions.

Additionally, we investigate this experiment from the perspective of student-teacher knowledge transfer to probe a deeper question: is matching the aggregate posterior sufficient to transfer a learned representation? We note that Huang & Wang (2017) has also previously utilized MMD regularization for representational knowledge distillation.

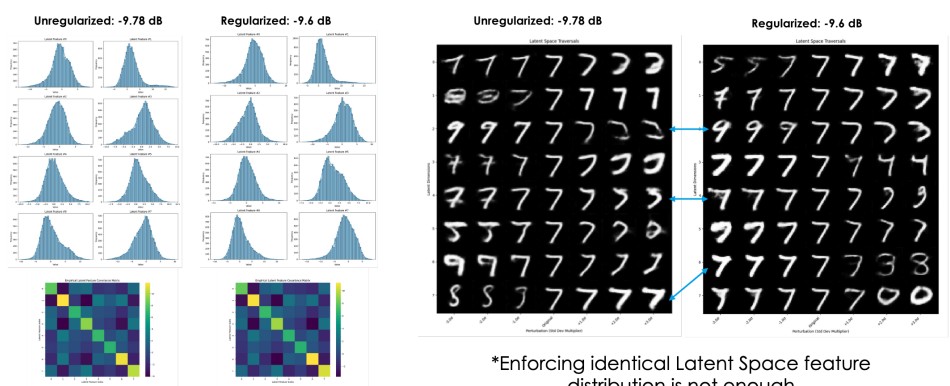

Figure 17: **Failure of Knowledge Transfer via Aggregate Posterior Matching.** (Left) Our MMD regularizer successfully forces a "student" model to replicate the latent distribution of a "teacher" model. (Right) Despite this, latent traversals show that the semantic meaning of the latent dimensions is completely different. For the same inputs (connected by lines), traversing a latent feature produces different visual changes, demonstrating that the underlying representations are not the same.

In this experiment, a "student" model was trained with our MMD regularizer to replicate the entire aggregate posterior distribution of a pre-trained, unregularized "teacher" autoencoder. As shown in the left panel of Figure 17, the student was highly successful—the learned marginals and covariance matrix are nearly identical to those of the teacher. However, the right panel reveals that this statistical alignment does not translate to functional equivalence. Latent traversals, which reveal the semantic meaning encoded in each dimension, produce wildly different outputs for the same input image. While a traversal in the teacher model might alter an attribute like the slant of a digit, the corresponding traversal in the student model results in a completely different transformation.

This finding is crucial: **matching the aggregate posterior distribution is not enough to transfer a model's learned representation**. Even with identical latent distributions, the two models have learned fundamentally different mappings from the input data to the latent space. This suggests that for true representational knowledge transfer, one must enforce a more complex constraint, such as matching the *joint* distribution of the latent representation with observable variables from the input data.

