# OpenReview forum: "Sculpting Latent Spaces With MMD: Disentanglement With Programmable Priors"
_ICLR.cc/2026/Conference — Submitted to ICLR 2026_

### Official Review · Reviewer_28ua · 2025-10-30

**Soundness:** 1
**Presentation:** 2
**Contribution:** 1
**Rating:** 2
**Confidence:** 5

**Summary:**

This manuscript presents a method to improve learning of disentangled latent space. The key component is using MMD instead of KL for regularization. The paper also proposes a metric LPS, latent predictability score.

**Strengths:**

1. The paper tries to address an important and open challenge.
2. The paper is written reasonably well, easy to follow.

**Weaknesses:**

The manuscript has the following issues:

1. It is well understood that MMD is better than KL in many applications. However the motivation in Fig 1, which empirically shows the proposed method learns better Gaussian latent space, is insufficient. Learning isotropic Gaussian latent space is closely related to learning disentangled latent space, but not the same. Especially for VAE and $\beta-$VAE, since latent space is subject to rotation, a more rigorous study is needed.
2. The paper is highly questionable in terms of execution. It's acceptable to propose a new metric, while propose a new method. But the new metric should be thoroughly examined before using it to champaign the proposed method. This manuscript failed to achieve this.
3. Missing an important reference. The proposed LPS is very similar to a recent paper Yeats et al, especially the concept of using reconstruction loss $d-1$ dimensions.
4. Lack of commonly used experimental benchmarks.

**Questions:**

1. Using a newly defined metric, LPS, to demonstrated the superiority of a proposed method needs to be executed thoroughly. The authors need to first establish the fact that the new metric is better than the existing metrics (e.g., MIG, DCI), in terms of consistency and robustness, across major existing disentanglement methods and datasets. I would encourage the authors to conduct such a comprehensive study before using the same metric to champaign the new method proposed by the same set of authors.

2. The details of LPS bear a close resemblance to a recent paper on disentanglement:

Yeats, E., Liu, F., Womble, D. and Li, H., 2022, October. Nashae: Disentangling representations through adversarial covariance minimization. In European Conference on Computer Vision (pp. 36-51). Cham: Springer Nature Switzerland.

which is not included in the reference section.

3. In LPS, "a regression model is trained to predict $z_i$", however there is no mentioning on how this model is trained, and more importantly, whether the quality of this model can be trusted to calculate LPS.  This is a much more nuanced usage of the regression model than in Yeats et al, where the purpose of the regression model is to **encourage** the disentanglement, not as a final yardstick to **evaluate the quality of disentanglement**. Again this brings up the question whether the LPS should be used to champaign the proposed method.

4. The experimental results are weak. As in Higgins et al, CelebA is commonly accepted as one of the baseline benchmarks. There is also a small datasets with known generating factors in Yeats et al. These benchmarks should be included in the manuscript.

---

> ### Author Response · Authors · 2025-11-21
>
> We thank the reviewer for their constructive remarks which we address one by one in the comments below. We also invite the reviewer to look at our summary of revisions (addressing all reviewers) in the top official comment above.
>
> * **1) It is well understood that MMD is better than KL in many applications. However the motivation in Fig 1, which empirically shows the proposed method learns better Gaussian latent space, is insufficient. Learning isotropic Gaussian latent space is closely related to learning disentangled latent space, but not the same. Especially for VAE and $\beta$-VAE, since latent space is subject to rotation, a more rigorous study is needed.**
>
>     We thank the reviewer for this excellent insight. We completely agree that "learning an isotropic Gaussian" and "learning a disentangled representation" are two distinct objectives. Our work is built on a two-part framework for disentanglement:
>
>     1.  Learning a representation (unsupervised) with **statistically independent** components (an ICA solution).
>     2.  **Aligning** this representation (semi-supervised) with semantically meaningful factors (to solve the NICA unidentifiability problem).
>
>     We then introduce the LPS score as a stable metric to *measure* (1) and subsequently reject the hypothesis that VAE and KL based based methods can learn representations that are mutually independent (as shown through a large LPS score in Fig 1 and Table 1). The reviewer's first point is perfectly illustrated by **Fig 1(d) (AE-KLD)**. While it enforces a diagonal covariance, its failure to shape the marginals (resulting in a near-perfectly-bad LPS of 0.97) demonstrates that it is *not* an isotropic Gaussian and, more importantly, *not* a statistically independent representation.
>
>     We clarify that Figure 1 is intended to demonstrate that the VAE and KL based methods (Fig 1a, 1b, 1d) fail to obtain a latent representation that is composed of mutually independent features. This is a critical failures since the main motivation for using VAE-based methods for disentanglement is based on its ability to learn a factorized posterior distribution. This unsupervised objective to learn mutually independent distribution is also the main objective of the NashAE.
>
>     The critical failure we observed in Fig 1 serves to motivate our use of MMD as a more powerful, non-parametric tool to successfully enforce (1), as shown in Fig 1(c), but also to introduce (2) through a semi-supervised mechanism. This is a key point of flexibility of our framework which previous baselines cannot address. This crucial two-part framing has been revised and expanded upon in the paper as we now additionally show how to systematically program the prior to ensure alignment with a semi-supervised mechanism.
>
> * **2) The paper is highly questionable in terms of execution. It's acceptable to propose a new metric, while propose a new method. But the new metric should be thoroughly examined before using it to champaign the proposed method. This manuscript failed to achieve this.**
>
>     We thank the reviewer for this valid concern, as any new metric requires thorough validation. We respectfully point the reviewer to **Appendix C (pages 16-18), "Metric Analysis,"** which is dedicated entirely to this examination.
>     In this section, we:
>
>     * Detail the powerful, non-linear regression models used: a Multi-Layer Perceptron (LPS-mlp) and LightGBM (LPS-lgbm).
>     * Demonstrate the critical **instability** of standard alignment metrics like DCI. As shown in **Fig 8(a)**, the variance across runs is often as large as the mean, making reliable comparison difficult (as noted by Locatello et al., 2019).
>     * Show that our LPS metric, in contrast, is exceptionally **stable** and **robust** across all models and datasets **(Fig 8b)**.
>     * Highlight a paradox (Fig 7) where models can achieve high DCI scores while being verifiably entangled (high LPS), reinforcing that alignment metrics are not a reliable proxy for independence.
>
>     This stability analysis is a core contribution, as it addresses a major flaw in current evaluation practices. We also remark that the use of the LPS score to measure (1) is partially validated through the NashAE's use of a core related concept for its own training objective.

---

> > ### Author Response · Authors · 2025-11-21
> >
> > * **3) Missing an important reference. The proposed LPS is very similar to a recent paper Yeats et al, especially the concept of using reconstruction loss dimensions.**
> >
> >     We thank the reviewer for this excellent reference which we were not aware of. We have updated our **contributions, Section 2, and Section 4.1** to explicitly credit and cite Yeats et al., and clarify the distinction. While NashAE uses a regression adversary as a *training objective* to enforce linear independence (covariance minimization), our LPS is a post-hoc *evaluation metric* using non-linear regressors (MLP/LGBM) to quantify intrinsic predictability. We additionally do provide comprehensive metric analysis of stability and robustness compared to classical supervised metrics in the Appendix. Furthermore, we clarify that our framework goes beyond independence (Part 1) to address alignment (Part 2) and latent space shaping (Part 3), which the NashAE does not address.
> >
> > * **4) Lack of commonly used experimental benchmarks.**
> >
> >     We thank the reviewer for this point. Our benchmark suite (dSprites, XY, XYC, XYCS, MNIST, CIFAR-10, and Tiny ImageNet) was chosen to test diverse properties: known ground-truth factors (dSprites, XY, XYC, XYCS), a different data modality (MNIST), and complex, high-dimensional real-world data (CIFAR, Tiny ImageNet). We believe this provides a comprehensive evaluation, but we agree that CelebA is another standard benchmark. As a result, we have added experiment results on CelebA in Table 1 and Table 6 which brings us to $8$ different datasets used to validate our framework.
> >
> > * **5) The experimental results are weak. As in Higgins et al, CelebA is commonly accepted as one of the baseline benchmarks. There is also a small datasets with known generating factors in Yeats et al. These benchmarks should be included in the manuscript.**
> >
> >     We thank the reviewer for this constructive suggestion. To address this, we have provided experimental results on the **CelebA (64x64)** dataset to Table 1 and Table 6 where we achieve state-of-the-art Latent Predictability Score (LPS), while maintaining high-quality reconstruction. We clarify that in total, we validate our framework on $5$ different synthetic datasets with known generating factors (dsprites, xy, xyc, xycs, CelebA), $2$ real world datasets (CIFAR10 and TinyImageNet, usually not included in disentanglement papers), and MNIST. We trust that this suite of benchmark datasets provides strong evidence to the scalability, stability, and robustness of our framework.

---

### Official Review · Reviewer_F6R6 · 2025-10-31

**Soundness:** 2
**Presentation:** 2
**Contribution:** 2
**Rating:** 2
**Confidence:** 3

**Summary:**

This paper presents a critique of the conventional Variational Autoencoder (VAE) framework for learning disentangled representations, arguing that its Kullback-Leibler (KL) divergence penalty is an unreliable mechanism for enforcing a factorized prior on the latent space. To address this, the authors introduce the Programmable Prior Framework, which replaces the KL divergence with a non-parametric Maximum Mean Discrepancy (MMD) regularizer. This MMD-based approach allows VAEs to sculpt the distribution of posterior for latent space which samples can be drawn from any target distribution, such as Gaussian, Uniform, or even Gaussian Mixture Models.
The authors further propose a novel unsupervised metric, the Latent Predictability Score (LPS), to quantify the mutual independence of latent features by measuring how well one latent dimension can be predicted from the others using a regression model. A lower score indicates greater independence.

**Strengths:**

1. The paper clearly identifies and provides compelling visual evidence (Figure 1) for a critical weakness in VAE-based disentanglement methods—the failure of the KL term to shape the aggregate posterior.
2. The proposal of the Latent Predictability Score (LPS) is a significant contribution. Its unsupervised nature makes it applicable to real-world datasets where ground-truth factors are unavailable. Furthermore, the authors convincingly demonstrate its superior stability compared to the high variance of alignment-based metrics like DCI, which is a crucial point for reliable model evaluation.

**Weaknesses:**

1. The paper's primary strength—the programmability of the prior—is also its main practical limitation. How to determine a proper target distribution for posterior remains unsolved. We can see the significant performance drop for choosing a wrong distribution from Table 12. The gaussian prior may not be the best one, but is robust for the most cases.
2. Due to the non-linearity of deep  neural networks, a gaussian distribution can be mapped to arbitrary distributions. Therefore, the posterior distribution is not a emergent problem in disentanglement learning.
3. From table 5, the proposed methods (AE-MMD) have lower MIG scores, low SAP, and so on. The methods did not exhibit strong disentangled representations on dSprites.

**Questions:**

Why AE-MMD has a high Covariance Ratio but low number of MIG or SAP?

---

> ### Author Response · Authors · 2025-11-21
>
> We thank the reviewer for their constructive remarks which we address one by one in the comments below. We also invite the reviewer to look at our summary of revisions (addressing all reviewers) in the top official comment above.
>
> * **1) The paper's primary strength—the programmability of the prior—is also its main practical limitation. How to determine a proper target distribution for posterior remains unsolved. We can see the significant performance drop for choosing a wrong distribution from Table 12.**
>
>     We thank the reviewer for this insightful comment and agree that optimal prior selection is a key challenge. Still, a key and meaningful finding of this paper is that the choice of the prior is a strong inductive bias in itself as the reviewer has correctly identified. To address this limitation, we replace this analysis in the main paper (moved to Appendix E) with a novel semi-supervised alignment mechanism (Section 3.1 and 4.4) which leverages some amount of labeled data to "program" the prior, effectively solving the alignment problem while maintaining mutual independence and the desired aggregate posterior distribution. We observe that peak alignment to all desired factors can be achieved with our framework by aligning only some of the semantic features. This ability demonstrates that our framework's ability to balance and address the $2$ key decoupled objectives of a disentangled representation enables semi-supervised recovery of generative factors.
>
> * **2) The gaussian prior may not be the best one, but is robust for the most cases. Due to the non-linearity of deep neural networks, a gaussian distribution can be mapped to arbitrary distributions. Therefore, the posterior distribution is not a emergent problem in disentanglement learning. From table 5, the proposed methods (AE-MMD) have lower MIG scores, low SAP, and so on. The methods did not exhibit strong disentangled representations on dSprites.**
>
>     This is a fantastic point, and it gets to the absolute core of our paper. The reviewer is exactly right: "a gaussian distribution can be mapped to arbitrary distributions." This is a succinct summary of the **NICA unidentifiability problem** (Khemakhem et al. 2020), which is the *key problem* we are addressing.
>     The reviewer notes our AE-MMD-Gaussian has low MIG/SAP scores while exhibiting very low LPS scores. We argue this is not a failure, but a **key finding**.
>     This result proves that **statistical independence and semantic alignment are two decoupled objectives of a disentangled representation**. Yet nearly all unsupervised methods for disentanglement utilize the learning of a mutually independent representation as the sole training objective.
>     The **Programmable Prior** provides a *solution* to this (e.g. ability to program the target prior and/or the semi-supervised alignment), providing a framework to inject the necessary inductive bias mentioned in Locatello 2019 needed to find the *aligned* independent solution. We have made this two-part "problem-and-solution" framing much clearer and compelling in our revisions.
>
> * **Q1) Why AE-MMD has a high Covariance Ratio but low number of MIG or SAP?**
>
>     The reviewer has precisely identified the **NICA unidentifiability problem**, which is the key motivation for our work.
>     Our paper frames disentanglement as two decoupled properties:
>
>     1.  **Statistical Independence:** Are the latent features $z_i$ mutually independent? We measure this with LPS (and CovRatio, though LPS is a stronger non-linear measure).
>     2.  **Semantic Alignment:** Does $z_1$ correspond to "x-position," $z_2$ to "shape," etc.? We measure this with MIG and SAP.
>
>     The result the reviewer points to (high CovRatio, low MIG) is not a failure, but a **key finding**: a model can achieve **perfect independence (solves Part 1) but have *zero* alignment (fails Part 2)**.
>     This happens because the model has found an *arbitrary* independent representation that is not the one the MIG/SAP metrics are looking for. For instance, the model may have learned to represent (x, y) position on a 45-degree tilted axis, or in spherical coordinates. These are perfectly valid, independent solutions, but MIG/SAP would (incorrectly) report a very low score.
>     This proves that statistical independence alone is not enough. This is why we introduce the **Programmable Prior** as a *solution* to inject the necessary bias that achieves alignment (Part 2).
>
> Lastly, we would like to respectfully clarify that our framework is separate from the VAE-based stochastic paradigm. In contrast to a VAE which stochastically aims to shape the posterior distribution of the latent space, we utilize a deterministic method to shape the **aggregate posterior distribution** of the latent space. In conclusion, our work cannot be summarized as replacing the KL penalty in a VAE with an MMD penalty.

---

> > ### Comment · Reviewer_F6R6 · 2025-11-28
> >
> > It makes sense for me in a semi-supervised setting. I would be happy to accept this if more advantages on semi-supervised settings are demonstrated. We encourage comparison between semi-supervised  disentanglement methods.

---

### Official Review · Reviewer_e3ck · 2025-11-01

**Soundness:** 3
**Presentation:** 2
**Contribution:** 2
**Rating:** 4
**Confidence:** 2

**Summary:**

In this paper, the authors tackle the limitations of KL-divergence-based VAE methods for learning disentangled representations. It introduces a flexible, architecture-agnostic framework using Maximum Mean Discrepancy to explicitly sculpt the latent space to match arbitrary priors, enabling what the authors term a programmable prior” The framework achieves state-of-the-art statistical independence of latent features without sacrificing reconstruction quality and provides a novel unsupervised metric, the Latent Predictability Score, to quantify disentanglement. Experiments across synthetic and real-world datasets demonstrate that MMD regularization can enforce both marginal and joint properties of latent distributions, allow alignment with interpretable features, and scale across varying latent dimensions.

**Strengths:**

This paper makes a significant contribution to the field of disentangled representation learning by directly addressing the limitations of KL-divergence-based VAE regularization. The proposed MMD-based framework is architecture-agnostic and provides a flexible, sample-based mechanism for sculpting the latent space to match arbitrary priors. This programmability is a clear strength, allowing practitioners to inject task-specific inductive biases into the representation, which is demonstrated through strong empirical results on complex datasets like CIFAR-10 and TinyImageNet. The framework also achieves state-of-the-art statistical independence of latent features, as measured by the novel Latent Predictability Score, without sacrificing reconstruction quality—a common trade-off in prior work.

Additionally, the introduction of the unsupervised LPS metric is an important methodological advance. By evaluating mutual independence without relying on ground-truth factors, LPS provides a robust and widely applicable tool for quantifying disentanglement. The experiments convincingly demonstrate that MMD regularization consistently enforces true statistical independence across diverse datasets and latent dimensions, highlighting the scalability and generality of the approach. The visualization and latent space copying experiments further illustrate the precision and flexibility of the programmable prior framework.

**Weaknesses:**

Despite its strengths, the framework has limitations in real-world applicability due to the challenge of selecting an optimal prior. While simple priors like factorized Gaussians are effective for achieving statistical independence, engineering priors that align with semantically meaningful features often requires domain knowledge that may not be available. This limits the ease of deploying the method in fully unsupervised scenarios where the underlying generative factors are unknown.
Another potential weakness lies in the computational complexity of MMD regularization in high-dimensional latent spaces. Although the paper demonstrates strong empirical performance, fitting complex priors may require careful kernel selection and tuning, which could hinder scalability or reproducibility for very large datasets. Furthermore, while the framework excels at matching marginal and aggregate distributions, it does not guarantee learning identical latent representations across models, which may limit its effectiveness in applications like knowledge distillation or causal representation learning

**Questions:**

How sensitive is the framework’s performance to the choice of kernel in the MMD regularizer, especially for high-dimensional latent spaces?

---

> ### Author Response · Authors · 2025-11-21
>
> We thank the reviewer for their constructive remarks which we address one by one in the comments below. We also invite the reviewer to look at our summary of revisions (addressing all reviewers) in the top official comment above.
>
> * **1) Despite its strengths, the framework has limitations in real-world applicability due to the challenge of selecting an optimal prior. While simple priors like factorized Gaussians are effective for achieving statistical independence, engineering priors that align with semantically meaningful features often requires domain knowledge that may not be available. This limits the ease of deploying the method in fully unsupervised scenarios where the underlying generative factors are unknown.**
>
>     We thank the reviewer for this insightful comment. We agree that purely unsupervised *alignment* is an ill-posed problem. However, we respectfully argue that our framework offers distinct, superior advantages in fully unsupervised scenarios even when domain knowledge is unavailable:
>
>     1.  **Achieving True Mutual Independence:** Unlike VAE-based frameworks, which often fail to enforce the prior on the aggregate posterior (leading to entanglement, see Fig 1 and Table 1), our MMD-based sculpting regularizer achieves SOTA statistical independence (low LPS) in a fully unsupervised setting.
>     2.  **Distribution Shaping:** By strictly enforcing the aggregate posterior to match a desired distribution (e.g., Gaussian or Uniform), we ensure the latent space is immediately compatible with downstream tasks (e.g., compression with quantization, see Fig 6).
>
>     Regarding **Semantic Alignment**, the reviewer correctly identifies that domain knowledge is required. This is consistent with the NICA unidentifiability results (Locatello et al. 2019; Khemakhem et al. 2020), which prove that unsupervised alignment is theoretically impossible without inductive bias. Our framework is designed to solve this specific deadlock in two ways. Firstly, we demonstrate that choosing the "correct" prior **ahead** of training can substantially improve alignment to semantic factors. Moreover, we introduce a semi-supervised mechanism/or learned prior scheme (Sec 4.4) that allows practitioners to inject available domain knowledge during training, aligning features *while simultaneously* maintaining the mutual independence and aggregate posterior shape required for the unsupervised goals above. Thus, the need for domain knowledge is not a limitation of our method, but a requirement of the alignment problem itself, which our framework uniquely accommodates.
>
> * **2) Another potential weakness lies in the computational complexity of MMD regularization in high-dimensional latent spaces. Although the paper demonstrates strong empirical performance, fitting complex priors may require careful kernel selection and tuning, which could hinder scalability or reproducibility for very large datasets.**
>
>     This is a very valid point. The reviewer is correct that the MMD regularizer has a $O(n^2)$ complexity with respect to the batch size $n$.
>     We argue this is a **necessary trade-off**. As we show in **Figure 1**, the $O(n)$ KL-based regularizers (Fig 1a, 1b) and even an aggregate KL (Fig 1d) *fundamentally fail* to solve the problem. They do not enforce the target prior, leading to highly entangled representations (e.g., LPS 0.97 for AE-KLD). On the flip side, we demonstrate our MMD method actually *works*.
>     Furthermore, this $O(n^2)$ cost is still highly competitive. It is significantly more stable and often more efficient than adversarial approaches (like FactorVAE) or iterative adversarial methods (like NashAE, which requires $k=5$ regressor updates *per batch* and to pass the batch through a deep MLP).
>     Finally, to address the reviewer's excellent point about **kernel tuning**, we explicitly designed our framework to be robust. As noted in **Appendix A.3**, we use two standard techniques: (1) a **mixture of RBF kernels** to capture multi-scale differences, and (2) the **median heuristic** to adaptively set the kernel bandwidth. This combination makes our method robust and removes the need for manual kernel-tuning, which is what enabled our stable results.
>     Still, we dedicated Appendix D to ablation studies on the scalability, stability and robustness of our MMD-based sculpting regularizer. We find that we achieve consistent results across many latent space dimensions, training batch sizes, regularization coefficients, number of bandwidths, and kernel types.

---

> > ### Author Response · Authors · 2025-11-21
> >
> > * **3) Furthermore, while the framework excels at matching marginal and aggregate distributions, it does not guarantee learning identical latent representations across models, which may limit its effectiveness in applications like knowledge distillation or causal representation learning**
> >
> >     We are thrilled the reviewer noted this finding from our **Appendix F.1 (Fig 17)**, as we consider it a very important and subtle result.
> >     The reviewer is exactly right: matching the aggregate posterior $q(z)$ is **not sufficient** to transfer the semantic representation $q(z|x)$.
> >     This finding perfectly reinforces our paper's main thesis. It provides further proof that the aggregate posterior $q(z)$ (the "shape" of the latent space) and the encoder's mapping $q(z|x)$ (the "meaning" of the latents) are two **decoupled components**.
> >     Our paper provides the foundational tool to fully control the first part, $q(z)$, in an unsupervised manner. As the reviewer correctly intuits, achieving full identifiability (e.g., for causal learning) requires (semi-)supervision. For this reason, we now show how to integrate a semi-supervised mechanism into the framework to also guarantee alignment to desired semantic features.
> >
> > * **Q1) How sensitive is the framework’s performance to the choice of kernel in the MMD regularizer, especially for high-dimensional latent spaces?**
> >
> >     This is an excellent question, which we also touched on in our response to Weakness 2. As detailed in **Appendix A.3**, we explicitly designed the regularizer to be **robust and insensitive** to manual tuning.
> >     We use two standard techniques from the MMD literature:
> >
> >     1.  A **mixture of RBF kernels** (with multipliers [0.5, 1.0, 2.0]), which allows the MMD to capture distributional discrepancies across multiple scales simultaneously.
> >     2.  The **median heuristic**, which adaptively sets the kernel bandwidth at each training step based on the batch statistics. This automatically scales the kernel, removing the need for a manually-tuned hyperparameter.
> >
> >     This combination makes the framework highly robust, which is what allowed us to achieve such stable results across all datasets without manual kernel-tuning.
> >     Still, we dedicated Appendix D to ablation studies on the scalability, stability and robustness of our MMD-based sculpting regularizer. We find that we achieve consistent results across many latent space dimensions, training batch sizes, regularization coefficients, number of bandwidths, and kernel types.

---

### Official Review · Reviewer_NjmS · 2025-11-03

**Soundness:** 2
**Presentation:** 2
**Contribution:** 1
**Rating:** 2
**Confidence:** 3

**Summary:**

This paper proposes a novel framework for learning disentangled representations by replacing the traditional KL divergence penalty in VAEs with the Maximum Mean Discrepancy (MMD). The authors argue that the per-sample KL-divergence is an unreliable mechanism for enforcing the desired factorized structure on the aggregate posterior distribution. By leveraging MMD, the proposed Programmable Prior Framework is claimed to be architecture-agnostic and non-parametric, allowing practitioners to "sculpt" the latent space to match any target prior distribution. The method demonstrates disentanglement performance on datasets like CIFAR-10 and Tiny ImageNet without the reconstruction quality trade-off common in $\beta$-VAE. Additionally, the authors introduce Latent Predictability Score (LPS), a unsupervised metric for quantifying mutual independence.

**Strengths:**

Clarity: Overall the paper is clearly written.

Novel Unsupervised Metric: The Latent Predictability Score (LPS) offers a new tool for quantifying mutual independence without relying on ground-truth factor labels.

**Weaknesses:**

Limited Theoretical Grounding & Novelty: The core technical idea is closely related to a lot of VAE variants e.g. WAE framework. The paper also need to formally justify the claim that $L_{ours}$ (Eq 5) is a lower bound on the log-likelihood.

Unsupported Claim Regarding Prior Flexibility: The paper claims that the proposed method supports "any prior", but the standard VAE objective is also theoretically valid for any analytic prior. The paper's strength is that MMD is empirically and practically more effective at sculpting the aggregate posterior to a non-Gaussian geometry due to its non-parametric nature. This distinction must be made clearer.

MMD Implementation Challenges: MMD is sensitive to the choice and tuning of the kernel function. A discussion or experiment on the robustness of the results to kernel choice would strengthen the practical utility.

Reproducibility: The provided code link is empty.

**Questions:**

Can the authors provide a rigorous theoretical derivation or reference for the claim that the objective ${L}_{ours}$ (Eq 5) is a lower bound on the log-likelihood?

MMD performance can be sensitive to the kernel function (e.g., Gaussian RBF) and its parameters. Could the authors include a robustness study?

---

> ### Author Response · Authors · 2025-11-21
>
> We thank the reviewer for their constructive remarks which we address one by one in the comments below. We also invite the reviewer to look at our summary of revisions (addressing all reviewers) in the top official comment above.
>
> * **1) Limited Theoretical Grounding and Novelty: The core technical idea is closely related to a lot of VAE variants e.g. WAE framework. The paper also need to formally justify the claim that (Eq 5) is a lower bound on the log-likelihood.**
>
>     We thank the reviewer for these important points, which allow us to clarify our novelty and theoretical grounding.
>     **On the Objective (Eq 5):** The reviewer is correct to ask for justification. Our objective, is a lower bound on the marginal log-likelihood because our model is deterministic and the MMD is a non-negative divergence, meaning we are optimizing the log-likelihood minus a non-negative penalty. This is a standard regularized objective (because we utilize a deterministic framework), not a VAE-style ELBO, and we have clarified this distinction in the text.
>
>     **On Novelty (vs. VAE/WAE):** Our novelty lies in our *target* and *purpose*.
>
>     * **vs. VAE:** A VAE stochastically regularizes the per-sample posterior $q(z|x)$. Our framework deterministically regularizes the **aggregate posterior** $q(z) = \mathbb{E}_{p(x)}[q(z|x)]$. As our **Figure 1** empirically proves, the VAE's objective *fails* to control the aggregate posterior, which is the central motivation for our direct approach. Moreover, we also provide an integrated semi-supervised mechanism for aligning to desired semantic factors.
>     * **vs. WAE:** WAE (Tolstikhin et al.) brilliantly used MMD for *generative modeling* (a $p(x)$ problem). We re-purpose MMD for a different task: *representation engineering* (a $q(z)$ problem). This novel application as a "Programmable Prior" to sculpt $q(z)$ for NICA and disentanglement is the core of our contribution.
>
> * **2) Unsupported Claim Regarding Prior Flexibility: The paper claims that the proposed method supports "any prior", but the standard VAE objective is also theoretically valid for any analytic prior. The paper's strength is that MMD is empirically and practically more effective at sculpting the aggregate posterior to a non-Gaussian geometry due to its non-parametric nature. This distinction must be made clearer.**
>
>     The reviewer is theoretically correct, and this distinction is precisely the one we aimed to make. The key difference is **analytic vs. sample-based**.
>     A KL-based regularizer is limited to priors with an *analytic* density function (e.g., Gaussian, Exponential, etc.).
>     Our MMD framework is **sample-based**. This means it can force the latent space to match *any* target distribution we can draw samples from, even those with no analytic form. This is a profound practical advantage. For example, as shown in our "latent space copying" experiment **(Figure 3)**, our framework successfully replicates the complex, co-dependent, non-analytic latent distribution from an unconstrained "teacher" AE. This task is **impossible** for a KL-based regularizer. This true "programmability" is a key tool for NICA and representation engineering.
>     Still, we aimed to further validate our claim that our sculpting framework is able to shape the latent distribution. To do this, we included a downstream training-ware quantization task (see Section 4.5) where our latent space must match the expected distribution of the quantizer. We observe that our framework provides the best results compared to baselines.

---

> ### Author Response · Authors · 2025-11-21
>
> * **3) MMD Implementation Challenges: MMD is sensitive to the choice and tuning of the kernel function. A discussion or experiment on the robustness of the results to kernel choice would strengthen the practical utility.**
>
>     This is an excellent question. We explicitly designed our framework to be **robust and insensitive** to manual kernel tuning. As detailed in **Appendix A.3**, we use two standard, robust techniques from the MMD literature:
>
>     1.  A **mixture of RBF kernels** (with multipliers [0.5, 1.0, 2.0]), which allows the MMD to capture distributional discrepancies across multiple scales simultaneously.
>     2.  The **median heuristic**, which adaptively sets the kernel bandwidth at each training step based on the batch's statistics. This automatically scales the kernel, removing the need for a manually-tuned hyperparameter.
>
>     This robust, standard setup is why our results are stable and reproducible without manual kernel-tuning, and we will highlight this in the main text.
>     Still, we dedicated Appendix D to ablation studies on the scalability, stability and robustness of our MMD-based sculpting regularizer. We find that we achieve consistent results across many latent space dimensions, training batch sizes, regularization coefficients, number of bandwidths, and kernel types.
>
> * **4) Reproducibility: The provided code link is empty.**
>
>     To strictly adhere to the double-blind policy, we are currently unable to release our code, as it is protected under a license that would compromise our anonymity. However, we can confirm that we intend to release the code upon acceptance.
>
> * **Q1) Can the authors provide a rigorous theoretical derivation or reference for the claim that the objective (Eq 5) is a lower bound on the log-likelihood?**
>
>     We thank the reviewer for this question. As stated in our response to (1), the objective of our deterministic framework in Eq 5, $L_{ours} = E[\log q_{\theta}(x)] - \lambda \cdot \text{MMD}^2(q(z),p(z))$, is a lower bound on the marginal log-likelihood $\mathbb{E}[\log q_{\theta}(x)]$. This is by definition, as we are optimizing the log-likelihood minus a non-negative penalty term ($\text{MMD}^2(P,Q) \ge 0$ for any P, Q). This is a standard regularized objective, and we have clarified this simple but important point in the revised text.
>
> * **Q2) MMD performance can be sensitive to the kernel function (e.g., Gaussian RBF) and its parameters. Could the authors include a robustness study?**
>
>     We agree that this is a critical aspect of practical utility. we dedicated Appendix D to ablation studies on the scalability, stability and robustness of our MMD-based sculpting regularizer. We find that we achieve consistent results across many latent space dimensions, training batch sizes, regularization coefficients, number of bandwidths, and kernel types.

---

### Author Response · Authors · 2025-11-21
**Summary of Changes and Edits (Phase 1 of discussion)**

In response to the reviewers' constructive feedback, we have revised the paper to clarify our theoretical positioning and demonstrate the practical utility of our method. Our core revision reframes the Programmable Prior Framework not just as an unsupervised  tool for disentanglement, but as a **Unified Representation Engineering Tool** that simultaneously addresses three distinct training objectives: (1) Statistical Independence, (2) Semantic Alignment, and (3) aggregate posterior distribution shaping (validated with training-aware quantization (QTA) experiments).

* **Reframed Contribution: A Unified 3-in-1 Framework (Abstract, Intro, Section 3):**
    * We have rewritten the introduction to frame disentanglement through the lens of **Nonlinear ICA (NICA)**, explicitly separating the goal into two distinct objectives: (1) unsupervised Independence and (2) semi-supervised Alignment.
    * *Addressing e3ck & F6R6:* We clarify that our unsupervised MMD objective (Eq. 5) solves Part 1 (Independence), while our new semi-supervised mechanism solves Part 2 (Alignment). All within the same Programmable Prior Framework (no additional regularization term etc.).
    * *Addressing 28ua & NjmS:* We clarified the theoretical distinction between our deterministic, regularized objective and the stochastic VAE ELBO.
    * **[See Edits]:** We invite reviewers to see the new "Contributions" list (https://prnt.sc/QqN_Nx9hB0rA) and the revised Section 3.1 ("Disentanglement and the Programmable Prior", https://prnt.sc/DSJbHSL0GvSN) which details this 3-part unification.

* **New Mechanism for Semantic Alignment (Section 3.1 and Section 4.4):**
    * *Addressing e3ck & F6R6 (Practical utility/Alignment):*
    * We introduced a semi-supervised mechanism that targets a joint distribution p(z, f(u)). This allows us to inject inductive bias (via pseudo-labels) to solve the NICA identifiability problem, ensuring alignment to semantically meaningful features while maintaining mutual independence.
    * **New Result (Fig 5):** We demonstrate that alignment (DCI) improves drastically as we align more pseudo-labels, confirming our framework can bridge the gap between unsupervised independence and semi-supervised interpretability. In fact, we observe that the peak alignment occurs with less than all pseudo labels testifying to our framework's ability to find mutually independent features.
    * **[See Edits]:** Please refer to the new **Section 3.1** (https://prnt.sc/DSJbHSL0GvSN) and **Section 4.4** (https://prnt.sc/SJdU5OLC6v_u) in the main text.

* **New Application: Validating Distribution Shaping with Out-of-the-Box Quantization (Section 4.5):**
    * *Strengthening Contribution (General Utility):*
    * We added experiments showing that by sculpting the aggregate posterior to match a quantizer's domain (e.g., Uniform or Gaussian), our model achieves approximatively 2-7 dB better reconstruction at low bit-rates compared to baselines.
    * **[See Edits]:** Please refer to the new **Section 4.5** (https://prnt.sc/W8PRiwQ56v9v) in the main text.

* **New Benchmark: CelebA Dataset (Table 1 & Table 6):**
    * *Addressing 28ua (Experimental weakness):*
    * We added the large-scale CelebA dataset to our main results (Table 1), demonstrating that our SOTA independence results hold on complex, real-world faces (LPS 0.08 vs around 0.4 baseline).
    * **[See Edits]:** New results are integrated into **Table 1** (Main Text) and detailed in **Table 6**.

* **Theoretical Clarifications & Related Work (Section 2 & 4.1):**
    * *Addressing 28ua (Yeats et al.) & NjmS (Lower Bound):*
    * We explicitly credit and cite **Yeats et al. (NashAE)** in our contributions and Section 4.1. We clarify that our LPS is an unsupervised *metric* for mutual independence (addressing stability/robustness in Appendix C), whereas theirs is a *training objective*.
    * We clarified in Section 3 that our objective (Eq. 5) is a standard regularized objective, distinct from the VAE's ELBO.

* **Stability & Robustness Analysis (Appendix D):**
    * *Addressing e3ck (Kernel tuning) & NjmS (Sensitivity):*
    * We added a comprehensive ablation study verifying that our MMD regularizer is robust to latent dimensionality, training batch size, regularization coefficient, bandwidths, and kernel choice.
    * **[See Edits]:** Please refer to the completely new **Appendix D** and **Figures 10, 11, 12**.

We trust that these comprehensive revisions fully address the reviewers' concerns, solidifying the novelty, utility, and consequence of our work for the field of representation learning. We respectfully invite the reviewers to re-evaluate our submission in light of these improvements.

To strictly adhere to the double-blind policy, we are currently unable to release our code, as it is protected under a license that would compromise our anonymity. However, we can confirm that we intend to release the code upon acceptance.

---

> ### Author Response · Authors · 2025-12-03
> **Additional Results and Experiments**
>
> We have further provided results to benchmark our method against the NashAE baseline (Yeats et al., ECCV 2022) (we ran all our experiments using their provided code at https://github.com/ericyeats/nashae-beamsynthesis and regularizing coefficient of $\lambda = 0.2$). As was pointed out by reviewer 28ua, this baseline is particularly relevant as it represents a class of unsupervised methods that enforce mutual independence via adversarial training of the predictability of the latent space—a mechanism conceptually similar to our LPS metric. While NashAE achieves reasonable LPS scores on some benchmarks, our method consistently outperforms it (see results here: https://prnt.sc/oD67jwxqmIfO), demonstrating superior robustness.
>
> Upon examining the adversary's $R^2$ plots (https://prnt.sc/3UoSoExY3ggX) which are consistent with results from their paper, we observed that the adversary failed to accurately fit the latent space. We know this because while the NashAE's trained adversary reports that it cannot predict the missing latent feature using all remaining features, our LPS score model can and reports substantial levels of entanglement. This highlights a common failure mode in adversarial methods where performance relies heavily on the adversary's capacity and training stability. Due to limited training iterations, the adversary could not sufficiently detect dependencies, resulting in residual entanglement that was detected by our LPS score. This finding reinforces the value of using LPS as a post-hoc metric to rigorously quantify disentanglement by way of lower-bounding the mutual dependence in the latent space after training, rather than relying solely on it as a training objective.
>
> We also extended our CelebA benchmarks to the $d=64$ latent dimension regime (complementing the standard $d=32$ experiments): https://prnt.sc/XX3WodJbD520. Our method demonstrated remarkable consistency, maintaining high performance even when the latent space was significantly overparameterized relative to the $\sim 40$ known factors. In stark contrast, baseline methods degraded significantly. Most notably, FactorVAE, which offered competitive performance at $d=32$ (LPS $\sim 0.07$), saw its LPS score worsen by a factor of $\sim 8\times$ at $d=64$. This dramatic degradation underscores the unreliability of VAE-based regularization in higher dimensions, whereas our framework remains robust.
>
> Regarding the semi-supervised mechanism, we are currently finalizing comparisons with established baselines, such as Locatello et al.'s Weakly-Supervised Disentanglement Without Compromises. We are confident our method is highly competitive; while Locatello et al. report DCI scores of $\sim 0.5$ on dSprites (see their Fig. 2), we achieve comparable performance using information from only two labels (see our Fig. 5) whereas their methods requires information on all labels (specifically, which features changed between sets of images). Furthermore, we conducted additional semi-supervised alignment experiments on MNIST and CelebA. By aligning the first latent dimension with the class label, we successfully forced the remaining dimensions to learn features strictly independent of class identity. We will provide latent traversals illustrating this clean separation in an appendix section of the final paper, contrasting it with the entangled outcomes of traditional VAE training on these datasets.
>
> These results confirm that our framework not only learns mutually independent features but also enables robust sculpting of the aggregate posterior for applications like compression, while offering unique flexibility for semi-supervised alignment. Crucially, while existing baselines typically optimize for a single aspect of disentanglement, our method establishes competitive performance (and in many cases new state-of-the-art) across all benchmarks and goals simultaneously.
>
> Furthermore, independent of the framework’s success, this work contributes significant novelty through its rigorous failure analysis (enabled through the LPS score) of dominant paradigms—specifically NashAE and VAEs—revealing their fundamental inability to enforce their core objective of mutual statistical independence. By demonstrating the necessity of the LPS score as a robust and stable metric for mutual independence we fill a long-standing gap in the literature: providing the field with a robust, unsupervised metric required to reliably evaluate mutual independence.

---

### Meta-Review · Area_Chair_ZLpD · 2026-01-07

**Summary:**

Reviewers raised concerns mainly about the clarity and positioning of the proposed method relative to existing approaches. In particular, Reviewer **NjmS** questioned the distinction between the proposed approach and prior work such as Wasserstein Autoencoders (WAE), which also use MMD for divergence computation and allow deterministic encoders. Reviewer **e3ck** raised questions about the choice of the target prior and the computational complexity of MMD, while Reviewer **28ua** noted that the evaluation relies on a self-proposed metric and requested clearer justification.

Overall, while the authors addressed several issues by adding additional experiments on CelebA and introducing a semi-supervised approach to learning the prior, the reviewers’ concerns highlighted the need for clearer differentiation from existing methods and stronger positioning of the contribution.

**Reviewer Concerns:**

### Addressed

1. **Expanded experimental evaluation:**
   The authors conducted additional experiments on CelebA, strengthening the empirical validation.

2. **Learning the target prior:**
   A new semi-supervised section was added to demonstrate how a suitable prior can be learned using pseudo-labels and neural networks.

### Outstanding

1. **Insufficient distinction from existing approaches:**
   The paper does not clearly distinguish its method from prior work, particularly Wasserstein Autoencoders (WAE), which also use MMD to measure divergence and allow deterministic encoders. The relationship and differences between the proposed approach and WAE remain insufficiently clarified.

**Reviewer Scores:**

Reviewer **NjmS** raised concerns about the relationship between the proposed method and prior work on WAE and VAE. While the authors briefly stated that WAE uses MMD for generative modeling whereas their work repurposes MMD for representation engineering, the distinction was not made sufficiently clear. As a result, this concern is not fully addressed, and the reviewer is likely to remain negative.

Reviewer **e3ck** questioned the choice of the target prior and the computational complexity of MMD. The authors argued that MMD-based distribution shaping is more suitable than VAE-based approaches when the prior is unknown, and provided an explanation of the computational cost. This response may partially address the concern, and the reviewer may remain negative or potentially raise their score.

Reviewer **F6R6** expressed concerns about how the target prior is chosen. In response, the authors added semi-supervised experiments demonstrating how to learn a suitable prior using pseudo-labels and neural networks. The reviewer showed interest in seeing additional results, suggesting they may remain negative or raise their score.

Reviewer **28ua** raised concerns that the paper primarily evaluates the method using a self-proposed metric. The authors added experiments analyzing this metric and argued that existing metrics can be unstable. This may partially address the concern, and the reviewer may remain negative or potentially raise their score.

---

### Decision · Program_Chairs · 2026-01-26

Reject